# On the Asymptotic Learning Curves of Kernel Ridge Regression under Power-law Decay

**Yicheng Li, Haobo Zhang**
Center for Statistical Science, Department of Industrial Engineering
Tsinghua University, Beijing, China
`{liyc22,zhang-hb21}@mails.tsinghua.edu.cn`

**Qian Lin** *
Center for Statistical Science, Department of Industrial Engineering
Tsinghua University, Beijing, China
`qianlin@tsinghua.edu.cn`

## Abstract

The widely observed 'benign overfitting phenomenon' in the neural network literature raises the challenge to the 'bias-variance trade-off' doctrine in the statistical learning theory. Since the generalization ability of the 'lazy trained' over-parametrized neural network can be well approximated by that of the neural tangent kernel regression, the curve of the excess risk (namely, the learning curve) of kernel ridge regression attracts increasing attention recently. However, most recent arguments on the learning curve are heuristic and are based on the 'Gaussian design' assumption. In this paper, under mild and more realistic assumptions, we rigorously provide a full characterization of the learning curve in the asymptotic sense under a power-law decay condition of the eigenvalues of the kernel and also the target function. The learning curve elaborates the effect and the interplay of the choice of the regularization parameter, the source condition and the noise. In particular, our results suggest that the 'benign overfitting phenomenon' exists in over-parametrized neural networks only when the noise level is small.

## 1 Introduction

Kernel methods, in particular kernel ridge regression (KRR), have been one of the most popular algorithms in machine learning. Its optimality under various settings has been an active topic since Caponnetto and De Vito [2007], Andreas Christmann [2008]. The renaissance of kernel methods arising from the neural tangent kernel (NTK) theory [Jacot et al., 2018], which shows that over-parametrized neural networks can be well approximated by certain kernel regression with the corresponding NTK, has posed further challenges about the interplay of generalization, regularization and noise level. For example, it has been observed empirically that over-parametrized neural networks can fit any data perfectly but also generalize well [Zhang et al., 2017], which contradicts to our traditional belief of bias-variance trade-off [Vapnik, 1999].

The aforementioned 'benign overfitting phenomenon' that overfitted neural networks generalize well attracts lots of attention recently. Researchers provide various explanations to reconcile the contradiction between it and the bias-variance trade-off principle. For example, Belkin et al. [2019] proposed the 'double descent theory' to explain why large model can generalize well; some other works (e.g., Liang and Rakhlin [2020]) argued that kernel interpolating estimators can generalize

---

*Qian Lin also affiliates with Beijing Academy of Artificial Intelligence, Beijing, China

37th Conference on Neural Information Processing Systems (NeurIPS 2023).

well in high dimensional settings. In contrast to the 'benign overfitting phenomenon', several other works (e.g., Rakhlin and Zhai [2018], Li et al. [2023a]) recently showed that kernel interpolation can not generalize in traditional fixed dimension setting. In order to understand the 'benign overfitting phenomenon', it would be of great interest to characterize the learning curve: the curve of the exact order of the generalization error of a certain algorithm (e.g., KRR) varying with respect to different choices of regularization parameters.

Recently, several works (e.g., Bordelon et al. [2020], Cui et al. [2021]) depicted the learning curve of KRR under the Gaussian design assumption that the eigenfunctions (see (5)) are i.i.d. Gaussian random functions. Though it is easy to figure out that the Gaussian design assumption can not be true in most scenarios, with some heuristic arguments, Cui et al. [2021] provide a description of the learning curves of KRR with respect to the regularization, source condition and noise levels. These works offered us some insights on the learning curve of KRR which strongly suggests that the learning curve should be U-shaped if the observations are noisy or monotone decreasing if the observations are noiseless.

In this paper, we consider the learning curves of KRR under the usual settings (without the Gaussian design assumption). Under mild assumptions, we rigorously prove the asymptotic rates of the excess risk, including both upper and lower bounds. These rates show the interplay of the eigenvalue decay of the kernel, the relative smoothness of the regression function, the noise and the choice of the regularization parameter. As a result, we obtain the traditional U-shaped learning curve for the noisy observation case and a monotone decreasing learning curve for the noiseless case, providing a full picture of the generalization of KRR in the asymptotic sense. Combined with the NTK theory, our results may also suggest that 'the benign overfitting phenomenon' may not exist if one trains a very wide neural network.

## 1.1   Our contributions

The main contribution of this paper is that we remove the unrealistic Gaussian design assumption in previous non-rigorous works [Bordelon et al., 2020, Cui et al., 2021] and provide mathematically solid proof of the exact asymptotic rates of KRR with matching upper and lower bounds.

To be precise, let us introduce the quantities $\lambda$, the regularization parameter in (1); $\beta$, the eigenvalue decay rate in (6), which characterizes the span of the underlying reproducing kernel Hilbert space (RKHS); and $s$, the smoothness index in (12), describes the relative smoothness of the regression function with respect to the RKHS. Here we note that larger $\beta$ implies better regularity the RKHS and also larger $s$ also implies better relative smoothness. Then, the asymptotic rates of the generalization error (excess risk) $R(\lambda)$ in the noisy case is roughly

$$R(\lambda) = \begin{cases} \Theta\big(\lambda^{\min(s,2)} + \sigma^2 \lambda^{-1/\beta}/n\big), & \text{if} \quad \lambda = \Omega(n^{-\beta}); \\ \Omega(\sigma^2), & \text{if} \quad \lambda = O(n^{-\beta}); \end{cases}$$

where $n$ is the number of the samples and $\sigma^2$ is the noise level. This result justifies the traditional U-shaped learning curve (see also Figure 1 on page 6) with respect to the regularization parameter.

For the technical part, we use the bias-variance decomposition and determine the exact rates of the both terms. Since the variance term was already considered in Li et al. [2023a], the main focus of this work is the bias term. Our technical contributions include:

- When the regularization parameter $\lambda$ is not so small, that is, $\lambda = \Omega(n^{-\beta})$, we provide sharp estimates of the asymptotic orders (Lemma 4.1) of the bias term with both upper and lower bounds. Our result holds for both the well-specified case ($s \geq 1$) and the mis-specified case ($s \in (0,1)$), which improves the upper bounds given in Zhang et al. [2023a].

- We further show an upper bound (Lemma A.12) of the bias term in the nearly interpolating case, i.e., $\lambda = O(n^{-\beta})$. The upper bound is tight and matches the information-theoretic lower bound provided in Proposition 4.4.

- Combining these results, we provide learning curves of KRR for both the noisy case (Theorem 3.2) and the noiseless case (Theorem 3.4). The results justify our traditional belief of the bias-variance trade-off principle.

- Our new techniques can also be generalized to other settings and might be of independent interest.

## 1.2 Related works

The optimality of kernel ridge regression has been studied extensively [Caponnetto and De Vito, 2007, Steinwart et al., 2009, Fischer and Steinwart, 2020, Zhang et al., 2023a]. Caponnetto and De Vito [2007] provided the classical optimality result of KRR in the well-specified case and the subsequent works further considered the mis-specified case. However, these works only provided an upper bound and the worst-case (minimax) lower bound, which are not sufficient for determining the precise learning curve. In order to answer the "benign overfitting" phenomenon [Bartlett et al., 2020, Liang and Rakhlin, 2020], several works [Rakhlin and Zhai, 2018, Buchholz, 2022, Beaglehole et al., 2022] tried to provide a lower bound for the kernel interpolation, which is a limiting case of KRR, but these works only focused on particular kernels and their techniques can hardly be generalized to provide a lower bound for KRR.

Another line of recent works considered the generalization performance of KRR under the Gaussian design assumption of the eigenfunctions [Bordelon et al., 2020, Jacot et al., 2020, Cui et al., 2021, Mallinar et al., 2022]. In particular, the learning curves of KRR was described in Bordelon et al. [2020], Cui et al. [2021], but heuristic arguments are also made in addition to the unrealistic Gaussian design assumption. Though the heuristic arguments are inspirational, a rigorous proof is indispensable if one plans to perform further investigations. In this work, we provide the first rigorous proof for most scenarios of the smoothness $s$, eigenvalue decay rate $\beta$, noise level $\sigma^2$ and the regularization parameter $\lambda$ based on the most common/realistic assumptions.

Recently, in order to show the so-called "saturation effect" in KRR, Li et al. [2023b] proved the exact asymptotic order of both the bias and the variance term when the regression function is very smooth and the regularization parameter $\lambda$ is relatively large. Inspired by their analysis, Li et al. [2023a] showed the exact orders of the variance term. Our work further determines the orders of the bias term, completing the full learning curve or KRR.

KRR is also connected with Gaussian process regression [Kanagawa et al., 2018]. Jin et al. [2021] claimed to establish the learning curves for Gaussian process regression and thus for KRR. However, as pointed out in Zhang et al. [2023b], there is a gap in their argument. Moreover, their results are also more restrictive than ours, see Section 3.3 for a comparison.

**Notations** We write $L^p(\mathcal{X}, \mathrm{d}\mu)$ for the Lebesgue space and sometimes abbreviate it as $L^p$. We use asymptotic notations $O(\cdot)$, $o(\cdot)$, $\Omega(\cdot)$ and $\Theta(\cdot)$, and use $\tilde{\Theta}(\cdot)$ to suppress logarithm terms. We also write $a_n \asymp b_n$ for $a_n = \Theta(b_n)$. We will also use the probability versions of the asymptotic notations such as $O_{\mathbb{P}}(\cdot)$. Moreover, to present the results more clearly, we denote $a_n = O^{\mathrm{poly}}(b_n)$ if $a_n = O(n^p b_n)$ for any $p > 0$, $a_n = \Omega^{\mathrm{poly}}(b_n)$ if $a_n = \Omega(n^{-p} b_n)$ for any $p > 0$, $a_n = \Theta^{\mathrm{poly}}(b_n)$ if $a_n = O^{\mathrm{poly}}(b_n)$, and $a_n = \Omega^{\mathrm{poly}}(b_n)$; and we add a subscript $_{\mathbb{P}}$ for their probability versions.

## 2 Preliminaries

Let $\mathcal{X} \subset \mathbb{R}^d$ be compact and $\rho$ be a probability measure on $\mathcal{X} \times \mathbb{R}$, whose marginal distribution on $\mathcal{X}$ is denoted by $\mu$. Suppose that we are given $n$ i.i.d. samples $(x_1, y_1), \ldots, (x_n, y_n)$ from $\rho$. Let $k$ be a continuous positive definite kernel $k$ over $\mathcal{X}$ and $\mathcal{H}$ be the separable reproducing kernel Hilbert space (RKHS) associated with $k$. Then, kernel ridge regression (KRR) obtains the regressor $\hat{f}_\lambda$ via the following convex optimization problem

$$\hat{f}_\lambda = \underset{f \in \mathcal{H}}{\arg\min} \left( \frac{1}{n} \sum_{i=1}^n (y_i - f(x_i))^2 + \lambda \|f\|_{\mathcal{H}}^2 \right), \tag{1}$$

where $\lambda > 0$ is the regularization parameter. Let us denote $X = (x_1, \ldots, x_n)$ and $\boldsymbol{y} = (y_1, \ldots, y_n)^T$. A closed form of (1) can be provided by the representer theorem [Andreas Christmann, 2008]:

$$\hat{f}_\lambda(x) = \mathbb{K}(x, X)(\mathbb{K}(X, X) + n\lambda)^{-1} \boldsymbol{y} \tag{2}$$

where $\mathbb{K}(x, X) = (k(x, x_1), \ldots, k(x, x_n))$ and $\mathbb{K}(X, X) = \big(k(x_i, x_j)\big)_{n \times n}$.

In terms of the generalization performance of $\hat{f}_\lambda$, we consider the excess risk with respect to the squared loss

$$\mathbb{E}_{x \sim \mu} \left[ \hat{f}_\lambda(x) - f_\rho^*(x) \right]^2 = \left\| \hat{f}_\lambda - f_\rho^* \right\|_{L^2(\mathcal{X}, \mathrm{d}\mu)}^2, \tag{3}$$

where $f_\rho^*(x) := \mathbb{E}_\rho[y \mid x]$ is the conditional expectation and is also referred to as the regression function. We aim to provide asymptotic orders of (3) with respect to $n$.

## 2.1 The integral operator

We will introduce the integral operator, which is crucial for the analysis, as the previous works [Caponnetto and De Vito, 2007, Lin et al., 2018]. Denote by $\mu$ the marginal probability measure of $\rho$ on $\mathcal{X}$. Since $k$ is continuous and $\mathcal{X}$ is compact, let us assume $\sup_{x \in \mathcal{X}} k(x, x) \le \kappa^2$. Then, it is known [Andreas Christmann, 2008, Steinwart and Scovel, 2012] that we have the natural embedding $S_\mu : \mathcal{H} \to L^2$, which is a Hilbert-Schmidt operator with Hilbert-Schmidt norm $\|S_\mu\|_{\mathrm{HS}} \le \kappa$. Let $S_\mu^* : L^2 \to \mathcal{H}$ be the adjoint operator of $S_\mu$ and $T = S_\mu S_\mu^* : L^2 \to L^2$. Then, it is easy to show that $T$ is an integral operator given by

$$(Tf)(x) = \int_{\mathcal{X}} k(x, y) f(y) \mathrm{d}\mu(y), \tag{4}$$

and it is self-adjoint, positive and trace-class (thus compact) with trace norm $\|T\|_1 \le \kappa^2$ [Caponnetto and De Vito, 2007, Steinwart and Scovel, 2012]. Moreover, the spectral theorem of compact self-adjoint operators and Mercer's theorem [Steinwart and Scovel, 2012] yield the decompositions

$$T = \sum_{i \in N} \lambda_i \langle \cdot, e_i \rangle_{L^2} e_i, \qquad k(x, y) = \sum_{i \in N} \lambda_i e_i(x) e_i(y), \tag{5}$$

where $N \subseteq \mathbb{N}$ is an index set, $\{\lambda_i\}_{i \in N}$ is the set of positive eigenvalues of $T$ in descending order, and $e_i$ is the corresponding eigenfunction. Furthermore, $\{e_i\}_{i \in N}$ forms an orthonormal basis of $\overline{\mathrm{Ran}\, S_\mu} \subseteq L^2$ and $\left\{ \lambda_i^{1/2} e_i \right\}_{i \in N}$ forms an orthonormal basis of $\overline{\mathrm{Ran}\, S_\mu^*} \subseteq \mathcal{H}$.

The eigenvalues $\lambda_i$ actually characterize the span of the RKHS and the interplay between $\mathcal{H}$ and $\mu$. Since we are interested in the infinite-dimensional case, we will assume $N = \mathbb{N}$ and assume the following polynomial eigenvalue decay as in the literature [Caponnetto and De Vito, 2007, Fischer and Steinwart, 2020, Li et al., 2023b], which is also referred to as the capacity condition or effective dimension condition. Larger $\beta$ implies better regularity of the functions in the RKHS.

**Assumption 1** (Eigenvalue decay). There is some $\beta > 1$ and constants $c_\beta, C_\beta > 0$ such that

$$c_\beta i^{-\beta} \le \lambda_i \le C_\beta i^{-\beta} \quad (i = 1, 2, \dots), \tag{6}$$

where $\lambda_i$ is the eigenvalue of $T$ defined in (5).

Such a polynomial decay is satisfied for the well-known Sobolev kernel [Fischer and Steinwart, 2020], Laplace kernel and, of most interest, neural tangent kernels for fully-connected multilayer neural networks [Bietti and Mairal, 2019, Bietti and Bach, 2020, Lai et al., 2023].

## 2.2 The embedding index of an RKHS

We will consider the embedding index of an RKHS to sharpen our analysis. Let us first define the fractional power $T^s : L^2 \to L^2$ for $s \ge 0$ by

$$T^s(f) = \sum_{i \in N} \lambda_i^s \langle f, e_i \rangle_{L^2} e_i. \tag{7}$$

Then, the interpolation space [Steinwart and Scovel, 2012, Fischer and Steinwart, 2020, Li et al., 2023b] $[\mathcal{H}]^s$ is define by

$$[\mathcal{H}]^s = \mathrm{Ran}\, T^{s/2} = \left\{ \sum_{i \in N} a_i \lambda_i^{s/2} e_i \ \Big| \ \sum_{i \in N} a_i^2 < \infty \right\} \subseteq L^2, \tag{8}$$

with the norm $\left\| \sum_{i \in N} a_i \lambda_i^{s/2} e_i \right\|_{[\mathcal{H}]^s} = \left( \sum_{i \in N} a_i^2 \right)^{1/2}$. One may easily verify that $[\mathcal{H}]^s$ is also a separable Hilbert space with an orthonormal basis $\left\{ \lambda_i^{s/2} e_i \right\}_{i \in N}$. Moreover, it is clear that $[\mathcal{H}]^0 = \overline{\operatorname{Ran} S_\mu} \subseteq L^2$ and $[\mathcal{H}]^1 = \overline{\operatorname{Ran} S_\mu^*} \subseteq \mathcal{H}$. It can also be shown that if $s_1 > s_2 \geq 0$, the inclusions $[\mathcal{H}]^{s_1} \hookrightarrow [\mathcal{H}]^{s_2}$ are compact [Steinwart and Scovel, 2012].

Now, we say $\mathcal{H}$ has an embedding property of order $\alpha \in (0, 1]$ if $[\mathcal{H}]^\alpha$ can be continuously embedded into $L^\infty(\mathcal{X}, \mathrm{d}\mu)$, that is, the operator norm

$$\| [\mathcal{H}]^\alpha \hookrightarrow L^\infty(\mathcal{X}, \mu) \| = M_\alpha < \infty. \tag{9}$$

Moreover, Fischer and Steinwart [2020, Theorem 9] shows that

$$\| [\mathcal{H}]^\alpha \hookrightarrow L^\infty(\mathcal{X}, \mu) \| = \left\| k_\mu^\alpha \right\|_{L^\infty} := \operatorname*{ess\,sup}_{x \in \mathcal{X}, \, \mu} \sum_{i \in N} \lambda_i^\alpha e_i(x)^2. \tag{10}$$

Therefore, since $\sup_{x \in \mathcal{X}} k(x, x) \leq \kappa^2$, we know that (9) always holds for $\alpha = 1$. By the inclusion relation of interpolation spaces, it is clear that if $\mathcal{H}$ has the embedding property of order $\alpha$, then it has the embedding properties of order $\alpha'$ for any $\alpha' \geq \alpha$. Consequently, we may introduce the following definition [Zhang et al., 2023b]:

**Definition 2.1.** The embedding index $\alpha_0$ of an RKHS $\mathcal{H}$ is defined by

$$\alpha_0 = \inf \left\{ \alpha : \| [\mathcal{H}]^\alpha \hookrightarrow L^\infty(\mathcal{X}, \mu) \| = M_\alpha < \infty \right\}. \tag{11}$$

It is shown in Fischer and Steinwart [2020, Lemma 10] that $\alpha_0 \geq \beta$ and we assume the equality holds as the following assumption.

**Assumption 2** (Embedding index). The embedding index $\alpha_0 = 1/\beta$, where $\beta$ is the eigenvalue decay in (6).

Lots of the usual RKHSs satisfy this embedding index condition. It is shown in Steinwart et al. [2009] that Assumption 2 holds if the eigenfunctions are uniformly bounded, namely $\sup_{i \in N} \|e_i\|_{L^\infty} < \infty$. Moreover, Assumption 2 also holds for the Sobolev RKHSs, RKHSs associated with periodic translation invariant kernels and RKHSs associated with dot-product kernels on spheres, see Zhang et al. [2023a, Section 4].

# 3 Main Results

Before presenting our main results, we have to introduce a source condition on the regression function. Since we will establish both precise learning rates, we have to characterize the exact smoothness order of $f_\rho^*$ rather than merely assume $f_\rho^*$ belongs to some interpolation space $[\mathcal{H}]^s$.

**Assumption 3** (Source condition). There are some $s > 0$ and a sequence $(a_i)_{i \geq 1}$ such that

$$f_\rho^* = \sum_{i=1}^\infty a_i \lambda_i^{s/2} i^{-1/2} e_i \tag{12}$$

and $0 < c \leq |a_i| \leq C$ for some constants $c, C$.

**Remark 3.1.** Assumption 3 is also considered in Cui et al. [2021, Eq. (8)] and a slightly weaker version of it is given in Jin et al. [2021, Assumption 5]. We only consider this simple form since there is no essential difference in the proof to consider the weaker version. From the definition (8) we can see that Assumption 3 implies $f_\rho^* \in [\mathcal{H}]^t$ for any $t < s$ but $f_\rho^* \notin [\mathcal{H}]^s$.

## 3.1 Noisy case

Let us first consider the noisy case with the following assumption:

**Assumption 4** (Noise). We assume

$$\mathbb{E}_{(x,y) \sim \rho} \left[ \left( y - f_\rho^*(x) \right)^2 \mid x \right] = \sigma^2 > 0, \quad \mu\text{-a.e. } x \in \mathcal{X}. \tag{13}$$

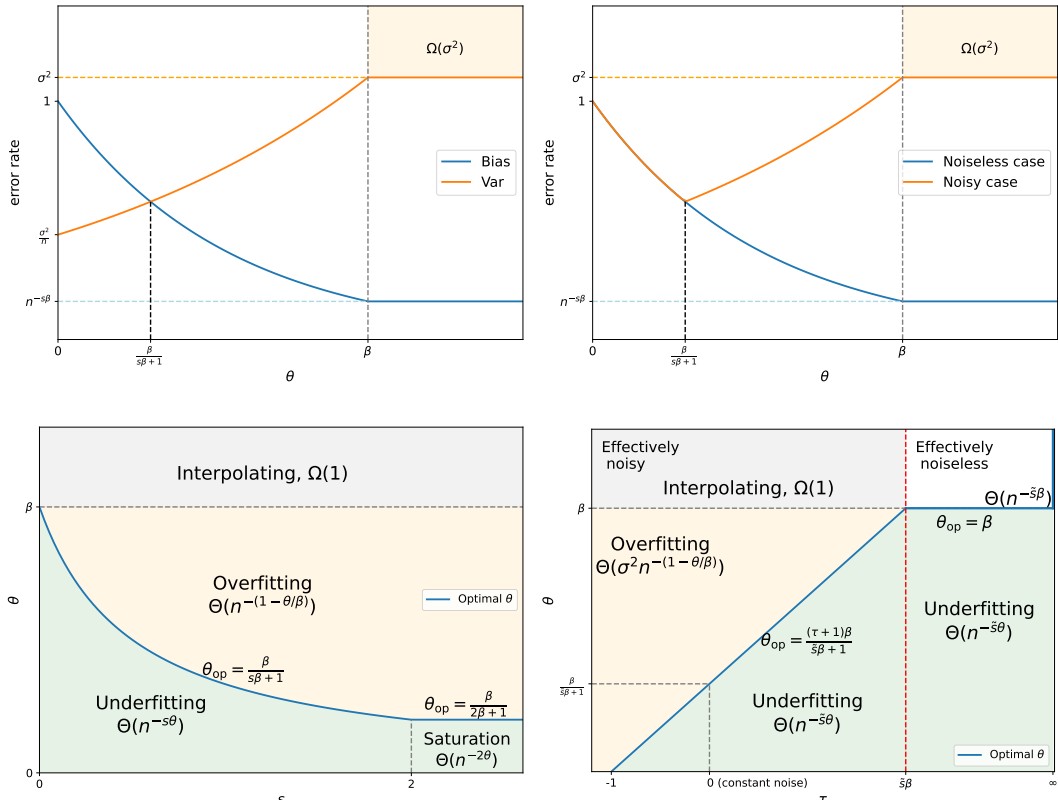

Figure 1: An illustration of the learning curves when choosing $\lambda = n^{-\theta}$. First row: The bias-variance plot and the error curves for the noisy and noiseless cases. Second row: Tow phase diagrams of the asymptotic rates of the excess risk with respect to parameter pairs $(\theta, s)$ and $(\theta, \tau)$, where we set $\sigma^2 = n^{-\tau}$ and $\tilde{s} = \min(s, 2)$. In the "underfitting" ("overfitting") region, bias (variance) is dominating. The "interpolating" region refers to the extreme cases of overfitting that the excess risk is lower bounded by a constant. For the first diagram we consider the case of constant noise. For the second diagram, the red vertical line shows the crossover of the noisy regime to the noiseless regime and an upper bound for the blank area on the upper-right corner is unknown yet.

For technical reason, we have to further assume the kernel to be Hölder-continuous, which is first in introduced in Li et al. [2023b]. This assumption is satisfied for the Laplace kernel, Sobolev kernels and neural tangent kernels.

**Assumption 5.** The kernel $k$ is Hölder-continuous, that is, there exists some $p \in (0, 1]$ and $L > 0$ such that

$$|k(x_1, x_2) - k(y_1, y_2)| \leq L\|(x_1, x_2) - (y_1, y_2)\|^p_{\mathbb{R}^{d \times d}}, \quad \forall x_1, x_2, y_1, y_2 \in \mathcal{X}. \tag{14}$$

**Theorem 3.2.** *Under Assumptions 1-5, suppose $\lambda \asymp n^{-\theta}$ for $\theta > 0$. Then,*

$$\mathbb{E}\left[\left\|\hat{f}_\lambda - f_\rho^*\right\|^2_{L^2} \;\Big|\; X\right] = \begin{cases} \tilde{\Theta}_{\mathbb{P}}\big(n^{-\min(s,2)\theta} + \sigma^2 n^{-(1-\theta/\beta)}\big), & \text{if} \quad \theta < \beta, \\ \Omega_{\mathbb{P}}^{\text{poly}}\big(\sigma^2\big), & \text{if} \quad \theta \geq \beta, \end{cases} \tag{15}$$

*where $\tilde{\Theta}_{\mathbb{P}}$ can be replaced with $\Theta_{\mathbb{P}}$ for the first case if $s \neq 2$.*

**Remark 3.3.** The two terms in the first case in Theorem 3.2 actually correspond to the bias and the variance term respectively. Balancing the two terms, we find the optimal regularization is $\theta_{\text{op}} = \frac{\beta}{\tilde{s}\beta+1}$ and the optimal rate is $\frac{\tilde{s}\beta}{\tilde{s}\beta+1}$, where $\tilde{s} = \min(s, 2)$, which recovers the classical optimal rate results [Caponnetto and De Vito, 2007]. Moreover, while we treat $\sigma^2$ as fixed for simplicity, we can also allow $\sigma^2$ to vary with $n$. Then, we can recover the results in Cui et al. [2021].

## 3.2 Noiseless case

**Theorem 3.4.** *Under Assumptions 1-3, assume further that the noise is zero, i.e., $y = f_\rho^*(x)$. Then, we have:*

- *Suppose $\lambda \asymp n^{-\theta}$ for $\theta \in (0, \beta)$, we have*

$$\mathbb{E}\left[\left\|\hat{f}_\lambda - f_\rho^*\right\|_{L^2}^2 \mid X\right] = \tilde{\Theta}_{\mathbb{P}}\left(n^{-\min(s, 2)\theta}\right), \tag{16}$$

*where $\tilde{\Theta}_{\mathbb{P}}$ can be replaced with $\Theta_{\mathbb{P}}$ if $s \neq 2$.*

- *Suppose $\lambda \asymp n^{-\theta}$ for $\theta \geq \beta$ and assume further that $s > 1$. Then,*

$$\mathbb{E}\left[\left\|\hat{f}_\lambda - f_\rho^*\right\|_{L^2}^2 \mid X\right] = O_{\mathbb{P}}^{\mathrm{poly}}\left(n^{-\min(s, 2)\beta}\right). \tag{17}$$

*Moreover, we have the information-theoretical lower rate:*

$$\sup_{\left\|f_\rho^*\right\|_{[\mathcal{H}]^s} \leq R} \mathbb{E}\left[\left\|\hat{f}_\lambda - f_\rho^*\right\|_{L^2}^2 \mid X\right] = \Omega(n^{-s\beta}), \tag{18}$$

*where $R > 0$ is a fixed constant.*

**Remark 3.5.** Theorem 3.4 shows that the generalization error of KRR in the noiseless case is monotone decreasing when $\theta$ increases and reaches the optimal rate $n^{-\beta}$ when $\theta \geq \beta$ if $s \leq 2$. Since the case $\theta \to \infty$ corresponds to kernel interpolation, our result implies that kernel interpolation is optimal when there is no noise. In contrast, as shown in Theorem 3.2 (or Li et al. [2023a]), kernel interpolation can not generalize in the noisy case. For the case $s > 2$, the KRR method suffers from saturation and the resulting convergence rate is limited to $n^{-2\beta}$, while the possible lower rate is $n^{-s\beta}$.

## 3.3 Discussion

Our results provide a full picture of the generalization of KRR, which is in accordance with our traditional belief of the bias-variance trade-off principle: the generalization error is a U-shaped curve with respect to the regularization parameter $\lambda$ in the noisy case and is monotone decreasing in the noiseless case. See Figure 1 on page 6 for an illustration.

Our rates coincide with the upper rates in the traditional KRR literature [Caponnetto and De Vito, 2007, Fischer and Steinwart, 2020]. Moreover, our results also recover the learning curves in Cui et al. [2021], but we do not need the strong assumption of Gaussian design eigenfunctions as in Cui et al. [2021], which may not be true in most cases. Our assumptions are mild and hold for a large class of kernels including the Sobolev kernels and the neural tangent kernels (NTK) on spheres.

Our results are based on the bias-variance decomposition and determining the rates for each term respectively. In the proof of Li et al. [2023b], they determined the rates of the variance term under the condition that $\theta < \frac{1}{2}$ and that of the bias term when $s \geq 2$ and $\theta < 1$. The subsequent work Li et al. [2023a] proved the rates of the variance term when $\theta < \beta$ and provided a near constant lower bound for $\theta \geq \beta$. Considering the counterpart, our works further prove the rates of the bias term, which finally enables us to determine the complete learning curve of KRR.

The connection between KRR and Gaussian process regression also results in the connection between their learning curves. Jin et al. [2021] claimed to show learning curves for Gaussian process regression. However, regardless of the gap in their proof as pointed out in Zhang et al. [2023b], their results are more restrictive than ours. Considering a boundedness assumption of the eigenfunctions that $\|e_i\|_\infty \leq Ci^\tau$ for some $\tau \geq 0$, they could only cover the regime of $\theta < \beta/(1 + 2\tau)$. Moreover, to approach the $\theta = \beta$ regime for the $\Omega(1)$ bound in the noisy case or the optimal rate in noiseless case, they have to require $\tau = 0$, that is, the eigenfunctions are uniformly bounded, but it is not true for some kernels such as dot-product kernels on spheres (and thus for NTK) since in general spherical harmonics are not uniformly bounded. In contrast, our embedding index assumption still holds in this case.

## 4 Proof sketch

We first introduce the following sample versions of the auxiliary integral operators, which are commonly used in the related literature [Caponnetto and De Vito, 2007, Fischer and Steinwart, 2020, Li et al., 2023b]. We define the sampling operator $K_x : \mathbb{R} \to \mathcal{H}$ by $K_x y = y k(x, \cdot)$, whose adjoint $K_x^* : \mathcal{H} \to \mathbb{R}$ is given by $K_x^* f = f(x)$. The sample covariance operator $T_X : \mathcal{H} \to \mathcal{H}$ is defined by

$$T_X := \frac{1}{n} \sum_{i=1}^{n} K_{x_i} K_{x_i}^*, \tag{19}$$

and the sample basis function is $g_Z := \frac{1}{n} \sum_{i=1}^{n} K_{x_i} y_i \in \mathcal{H}$. As shown in Caponnetto and De Vito [2007], the operator form of KRR writes

$$\hat{f}_\lambda = (T_X + \lambda)^{-1} g_Z. \tag{20}$$

Let us further define

$$\tilde{g}_Z := \mathbb{E}\left(g_Z | X\right) = \frac{1}{n} \sum_{i=1}^{n} K_{x_i} f_\rho^*(x_i) \in \mathcal{H}, \tag{21}$$

and

$$\tilde{f}_\lambda := \mathbb{E}\left(\hat{f}_\lambda | X\right) = (T_X + \lambda)^{-1} \tilde{g}_Z \in \mathcal{H}. \tag{22}$$

Then, the traditional bias-variance decomposition [Li et al., 2023b, Zhang et al., 2023a] yields

$$\mathbb{E}\left(\left\|\hat{f}_\lambda - f_\rho^*\right\|_{L^2}^2 \mid X\right) = \mathbf{Bias}^2(\lambda) + \mathbf{Var}(\lambda), \tag{23}$$

where

$$\mathbf{Bias}^2(\lambda) := \left\|\tilde{f}_\lambda - f_\rho^*\right\|_{L^2}^2, \quad \mathbf{Var}(\lambda) := \frac{\sigma^2}{n^2} \sum_{i=1}^{n} \left\|(T_X + \lambda)^{-1} k(x_i, \cdot)\right\|_{L^2}^2. \tag{24}$$

### 4.1 The noisy case

To prove the desired result, we have to establish the asymptotic orders of both $\mathbf{Bias}^2(\lambda)$ and $\mathbf{Var}(\lambda)$. We first prove the asymptotic order of $\mathbf{Bias}^2(\lambda)$ as one of our technical contributions. As far as we know, we are the first to provide such a lower bound in (25).

**Lemma 4.1.** *Under Assumptions 1,2,3, suppose $\lambda \asymp n^{-\theta}$ for $\theta \in (0, \beta)$. Then,*

$$\mathbf{Bias}^2(\lambda) = \tilde{\Theta}_{\mathbb{P}}\left(n^{-\min(s,2)\theta}\right), \tag{25}$$

*where $\tilde{\Theta}_{\mathbb{P}}$ can be replaced with $\Theta_{\mathbb{P}}$ if $s \neq 2$.*

*Proof sketch of Lemma* 4.1. Denote $\tilde{s} = \min(s, 2)$. We first introduce the regularized regression function $f_\lambda := T(T + \lambda)^{-1} f_\rho^*$ and triangle inequality implies

$$\mathbf{Bias}(\lambda) = \left\|\tilde{f}_\lambda - f_\rho^*\right\|_{L^2} \geq \left\|f_\lambda - f_\rho^*\right\|_{L^2} - \left\|\tilde{f}_\lambda - f_\lambda\right\|_{L^2}.$$

There is no randomness in the first term and we can use the expansion (12) and (5) to show that $\left\|f_\lambda - f_\rho^*\right\|_{L^2} = \tilde{\Theta}\left(n^{-\tilde{s}\theta}\right)$. Then, we have to prove the error term $\left\|\tilde{f}_\lambda - f_\lambda\right\|_{L^2}$ to be infinitesimal with respect to the main term, which is the main difficulty since it requires a refined analysis. Previous work only consider the case $\theta = \frac{\beta}{\tilde{s}\beta+1}$ (corresponding to the optimal regularization) and show an $O(n^{-\tilde{s}\theta})$ bound rather than the $o(n^{-\tilde{s}\theta})$ bound that we require. For the proof, we (1) apply the concentration techniques in Fischer and Steinwart [2020]; (2) consider the $L^q$-embedding property in Zhang et al. [2023a] for the mis-specified case when $s$ is small; (3) sharpen the estimation by exploiting the embedding property $\alpha_0 = 1/\beta$ and $\theta < \beta$. For the detail, see Section 2.2 in the supplementary material. □

The variance term has been analyzed in Li et al. [2023a]. We present the following proposition as a combination of Proposition 5.3 and Theorem 5.10 in Li et al. [2023a].

**Proposition 4.2.** Under Assumptions 1-5, suppose that $\lambda \asymp n^{-\theta}$. Then,

$$\mathbf{Var}(\lambda) = \begin{cases} \Theta_{\mathbb{P}}^{\mathrm{poly}}\big(\sigma^2 n^{-(1-\theta/\beta)}\big), & \text{if } \theta < \beta; \\ \Omega_{\mathbb{P}}^{\mathrm{poly}}\big(\sigma^2\big), & \text{if } \theta \geq \beta. \end{cases} \tag{26}$$

## 4.2 The noiseless case

For the noiseless case, the variance term vanishes in (23), and thus we only need to consider the bias term. Since we have already established the estimation for large $\lambda$ in Lemma 4.1, we focus on the case of small $\lambda$.

**Lemma 4.3.** *Under Assumptions 1,2,3, assume further $s > 1$. Suppose $\lambda \asymp n^{-\theta}$ for $\theta \geq \beta$. Then,*

$$\mathbf{Bias}^2(\lambda) = O_{\mathbb{P}}^{\mathrm{poly}}(n^{-\min(s,2)\beta}). \tag{27}$$

*Proof sketch of Lemma A.12.* Intuitively, we hope to bound $\mathbf{Bias}^2(\lambda)$ with $\mathbf{Bias}^2(\tilde{\lambda})$ for $\tilde{\lambda} > \lambda$ such that concentration still works. However, we can not directly derive no monotone property of $\mathbf{Bias}(\lambda)$. Nevertheless, since $f_\rho^* \in \mathcal{H}$ when $s > 1$, the bias term can be written as

$$\mathbf{Bias}(\lambda) = \left\| \lambda(T_X + \lambda)^{-1} f_\rho^* \right\|_{L^2} = \left\| T^{\frac{1}{2}} \lambda(T_X + \lambda)^{-1} f_\rho^* \right\|_{\mathcal{H}} \leq \left\| T^{\frac{1}{2}} \lambda(T_X + \lambda)^{-1} \right\|_{\mathscr{B}(\mathcal{H})} \left\| f_\rho^* \right\|_{\mathcal{H}}.$$

Then, by operator calculus we can show that

$$\left\| T^s \left[ \lambda(T_X + \lambda)^{-1} \right] \right\|_{\mathscr{B}(\mathcal{H})} \leq \left\| T^s \left[ \tilde{\lambda}(T_X + \tilde{\lambda})^{-1} \right] \right\|_{\mathscr{B}(\mathcal{H})}$$

reducing $\lambda$ to $\tilde{\lambda}$. Now, we can replace $T_X$ with $T$ using concentration results and derive the desired upper bound. □

The following proposition shows that the upper bound in Lemma A.12 matches the information-theoretical lower bound. The proof follows idea of the minimax principle [Micchelli and Wahba, 1979] and is deferred to the supplementary material.

**Proposition 4.4.** Suppose Assumption 1 holds and $s \geq 1$. For any $X = (x_1, \ldots, x_n)$, we have

$$\sup_{\|f_\rho^*\|_{[\mathcal{H}]^s} \leq R} \mathbf{Bias}^2(\lambda) = \Omega\big(n^{-s\beta}\big), \tag{28}$$

where we note that here $\mathbf{Bias}(\lambda)$ is viewed as a function depending also on $f_\rho^*$ and $X$.

## 5 Experiments

Lots of numerical experiments on both synthetic data and real data are done to study to learning curves of KRR [Li et al., 2023b, Cui et al., 2021]. In this section, we consider numerical experiments on a toy model to verify our theory.

Let us consider the kernel $k(x, y) = \min(x, y)$ and $x \sim \mathcal{U}[0, 1]$. Then, the corresponding RKHS is [Wainwright, 2019]

$$\mathcal{H} = \left\{ f : [0, 1] \to \mathbb{R} \,\middle|\, f \text{ is absolutely continuous, } f(0) = 0, \int_0^1 (f'(x))^2 \mathrm{d}x < \infty \right\}$$

and the eigenvalue decay rate $\beta = 2$. Moreover, the eigensystem of $k$ is known to be $\lambda_i = \left(\frac{2i-1}{2}\pi\right)^{-2}$ and $e_i(x) = \sqrt{2}\sin\left(\frac{2i-1}{2}\pi x\right)$, which allows us to directly compute the smoothness of certain functions. For some $f^*$, we generate data from the model $y = f^*(x) + \varepsilon$ where $\varepsilon \sim \mathcal{N}(0, 0.05)$ and perform KRR with $\lambda = cn^{-\theta}$ for different $\theta$'s with some fixed constant $c$. Then, we numerically compute the variance, bias and excess risk by Simpson's formula with $N \gg n$ nodes. Repeating the experiment for $n$ ranged in 1000 to 5000, we can estimate the convergence rate $r$ by a logarithmic least-squares $\log \mathrm{err} = r \log n + b$ on the values (variance, bias and excess risk). The results are collected in Table 1 on page 10. It can be seen that the resulting values basically match the theoretical values and we conclude that our theory is supported by the experiments. For more experiments and more details, we refer to the supplementary material.

| | $f^*(x) =$ | $\cos 2\pi x\ (s = \frac{1}{2})$ | | $\sin 2\pi x\ (s = 1.5)$ | | $\sin \frac{3}{2}\pi x\ (s = \infty)$ | |
|---|---|---|---|---|---|---|---|
| $\theta$ | Variance | Bias | Risk | Bias | Risk | Bias | Risk |
| 0.2 | 0.90 (0.90) | 0.13 (0.10) | 0.13 (0.10) | 0.34 (0.30) | 0.34 (0.30) | 0.40 (0.40) | 0.42 (0.40) |
| 0.4 | 0.80 (0.80) | 0.22 (0.20) | 0.22 (0.20) | 0.68 (0.60) | 0.69 (0.60) | 0.82 (0.80) | **0.81 (0.80)** |
| 0.5 | 0.75 (0.75) | 0.26 (0.25) | 0.26 (0.25) | 0.84 (0.75) | **0.79 (0.75)** | 1.04 (1.00) | 0.77 (0.75) |
| 1.0 | 0.49 (0.50) | 0.54 (0.50) | **0.52 (0.50)** | 1.69 (1.50) | 0.49 (0.50) | 2.21 (2.00) | 0.49 (0.50) |
| 2.0 | 0.00 (0.00) | 1.05 (1.00) | 0.09 (0.00) | 3.26 (3.00) | 0.00 (0.00) | 3.99 (4.00) | 0.00 (0.00) |
| 3.0 | 0.00 (0.00) | 1.05 (1.00) | 0.09 (0.00) | 3.26 (3.00) | 0.00 (0.00) | 3.98 (4.00) | 0.00 (0.00) |

Table 1: Asymptotic rates of bias, variance and excess risk under three regressions and different choices of $\theta$. The numbers in parenthess are the theoretical values. The bolded cells correspond to the best rate over the choices of $\theta$'s.

## 6 Conclusion

In this paper, we prove rigorously the learning curves of KRR, showing the interplay of the eigenvalue decay of the kernel, the relative smoothness of the regression function, the noise and the choice of the regularization parameter. The results justify our traditional bias-variance trade-off principle and provide a full picture of the generalization performance of KRR. These results will help us better understand the generalization mystery of neural networks.

As for future works, we notice that for the nearly interpolating regime when $\theta \geq \beta$, there are still some missing parts due to technical limitations. We expect that further analysis will prove the exact orders of the variance term like that given in Mallinar et al. [2022] under the Gaussian design assumption. We also hypothesize that Lemma A.12 still holds in the mis-specified case ($s < 1$).

## Acknowledgments and Disclosure of Funding

This work is supported in part by the Beijing Natural Science Foundation (Grant Z190001) and National Natural Science Foundation of China (Grant 11971257).

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

# A Detailed proofs

The first step of the proof is the traditional bias-variance decomposition. Let us further define

$$\tilde{g}_Z := \mathbb{E}\left(g_Z | X\right) = \frac{1}{n} \sum_{i=1}^{n} K_{x_i} f_\rho^*(x_i) \in \mathcal{H}, \tag{29}$$

and

$$\tilde{f}_\lambda := \mathbb{E}\left(\hat{f}_\lambda | X\right) = (T_X + \lambda)^{-1} \tilde{g}_Z \in \mathcal{H}. \tag{30}$$

Recalling (20), we have

$$\hat{f}_\lambda = \frac{1}{n}(T_X + \lambda)^{-1} \sum_{i=1}^{n} K_{x_i} y_i = \frac{1}{n}(T_X + \lambda)^{-1} \sum_{i=1}^{n} K_{x_i}(f_\rho^*(x_i) + \epsilon_i)$$

$$= (T_X + \lambda)^{-1} \tilde{g}_Z + \frac{1}{n} \sum_{i=1}^{n} (T_X + \lambda)^{-1} K_{x_i} \epsilon_i,$$

so that

$$\hat{f}_\lambda - f_\rho^* = \left(\tilde{f}_\lambda - f_\rho^*\right) + \frac{1}{n} \sum_{i=1}^{n} (T_X + \lambda)^{-1} K_{x_i} \epsilon_i.$$

Taking expectation over the noise $\epsilon$ conditioned on $X$, since $\varepsilon | x$ are independent noise with mean 0 and variance $\sigma^2$, we have

$$\mathbb{E}\left(\left\|\hat{f}_\lambda - f_\rho^*\right\|_{L^2}^2 \,\Big|\, X\right) = \mathbf{Bias}^2(\lambda) + \mathbf{Var}(\lambda), \tag{31}$$

where

$$\mathbf{Bias}^2(\lambda) := \left\|\tilde{f}_\lambda - f_\rho^*\right\|_{L^2}^2, \quad \mathbf{Var}(\lambda) := \frac{\sigma^2}{n^2} \sum_{i=1}^{n} \left\|(T_X + \lambda)^{-1} k(x_i, \cdot)\right\|_{L^2}^2. \tag{32}$$

## A.1 The variance term

**Theorem A.1.** *Under Assumptions 1-5, suppose that $\lambda \asymp n^{-\theta}$. Then,*

$$\mathbf{Var}(\lambda) = \begin{cases} \Theta_{\mathbb{P}}^{\mathrm{poly}}\left(\sigma^2 n^{-(1-\theta/\beta)}\right), & \text{if } \theta < \beta; \\ \Omega_{\mathbb{P}}^{\mathrm{poly}}\left(\sigma^2\right), & \text{if } \theta \geq \beta. \end{cases} \tag{33}$$

The computation in Li et al. [2023b] shows that

$$\mathbf{Var}(\lambda) = \frac{\sigma^2}{n^2} \int_{\mathcal{X}} \mathbb{K}(x, X)(K + \lambda)^{-2} \mathbb{K}(X, x) \mathrm{d}\mu(x).$$

Then, Theorem A.1 directly follows from Proposition 5.3 and Theorem 5.10 in Li et al. [2023a].

## A.2 The bias term

**Theorem A.2.** *Under Assumptions 1,2,3, suppose $\lambda \asymp n^{-\theta}$ for $\theta \in (0, \beta)$. Then,*

$$\mathbf{Bias}^2(\lambda) = \tilde{\Theta}_{\mathbb{P}}\left(n^{-\min(s, 2)\theta}\right), \tag{34}$$

*where $\tilde{\Theta}_{\mathbb{P}}$ can be replaced with $\Theta_{\mathbb{P}}$ if $s \neq 2$.*

Let us define the regularized version of the regression function

$$f_\lambda := (T + \lambda)^{-1} T f_\rho^*. \tag{35}$$

Then, the triangle inequality implies that

$$\mathbf{Bias}(\lambda) = \left\| \tilde{f}_\lambda - f_\rho^* \right\|_{L^2} \geq \left\| f_\lambda - f_\rho^* \right\|_{L^2} - \left\| \tilde{f}_\lambda - f_\lambda \right\|_{L^2} \tag{36}$$

Then, the proof of Theorem A.2 is the combination of the following Lemma A.3 (with $\gamma = 0$) and Lemma A.4, showing that the main term $\left\| f_\lambda - f_\rho^* \right\|_{L^2} = \tilde{\Theta}_{\mathbb{P}}\left( n^{-\min(s,2)\theta/2} \right)$ and the error term $\left\| \tilde{f}_\lambda - f_\lambda \right\|_{L^2} = o_{\mathbb{P}}\left( n^{-\min(s,2)\theta/2} \right)$.

**Lemma A.3.** *Under Assumptions 1 and 3, for any $0 \leq \gamma < s$, we have*

$$\left\| f_\lambda - f_\rho^* \right\|_{[\mathcal{H}]^\gamma}^2 \asymp \begin{cases} \lambda^{s-\gamma}, & s - \gamma < 2; \\ \lambda^2 \ln \frac{1}{\lambda}, & s - \gamma = 2; \\ \lambda^2, & s - \gamma > 2. \end{cases} \tag{37}$$

*Proof.* From the definition of interpolating norms, letting $p = (s - \gamma)/2$, we have

$$\left\| f_\lambda - f_\rho^* \right\|_{[\mathcal{H}]^\gamma}^2 = \sum_{i=1}^\infty a_i^2 \frac{\lambda^2}{(\lambda_i + \lambda)^2} (\lambda_i^s i^{-1}) \lambda_i^{-\gamma} \asymp \lambda^2 \sum_{i=1}^\infty \left( \frac{\lambda_i^p}{\lambda_i + \lambda} \right)^2 i^{-1}. \tag{38}$$

Then result then follows by applying Proposition B.2 for the last series. $\square$

The following lemma shows the error term in (36) is infinitesimal, whose proof relies on fine-grained concentration results established in Section A.3.

**Lemma A.4.** *Under Assumptions 1-3. Suppose $\lambda \asymp n^{-\theta}$ with $\theta \in (0, \beta)$, then*

$$\left\| \tilde{f}_\lambda - f_\lambda \right\|_{L^2} = o_{\mathbb{P}}\left( n^{-\min(s,2)\theta/2} \right) \tag{39}$$

*Proof.* We begin with

$$\begin{aligned} \left\| \tilde{f}_\lambda - f_\lambda \right\|_{L^2} &= \left\| T^{\frac{1}{2}} \left( \tilde{f}_\lambda - f_\lambda \right) \right\|_{\mathcal{H}} \\ &\leq \left\| T^{\frac{1}{2}} T_\lambda^{-\frac{1}{2}} \right\| \cdot \left\| T_\lambda^{\frac{1}{2}} T_{X\lambda}^{-1} T_\lambda^{\frac{1}{2}} \right\| \cdot \left\| T_\lambda^{-\frac{1}{2}} \left( \tilde{g}_Z - T_{X\lambda} f_\lambda \right) \right\|_{\mathcal{H}}. \end{aligned} \tag{40}$$

From operator calculus we know $\left\| T^{\frac{1}{2}} T_\lambda^{-\frac{1}{2}} \right\| \leq 1$. Moreover, since $\theta < \beta$ and the embedding index $\alpha_0 = 1/\beta$, by Lemma B.5 we get $\left\| T_\lambda^{\frac{1}{2}} T_{X\lambda}^{-1} T_\lambda^{\frac{1}{2}} \right\| \leq 3$ with high probability as long as $n$ is sufficiently large. For the last term in (40), we have

$$\begin{aligned} T_\lambda^{-\frac{1}{2}} \left( \tilde{g}_Z - T_{X\lambda} f_\lambda \right) &= T_\lambda^{-\frac{1}{2}} \left[ \left( \tilde{g}_Z - (T_X + \lambda + T - T) f_\lambda \right) \right] \\ &= T_\lambda^{-\frac{1}{2}} \left[ (\tilde{g}_Z - T_X f_\lambda) - (T + \lambda) f_\lambda + T f_\lambda \right] \\ &= T_\lambda^{-\frac{1}{2}} \left[ (\tilde{g}_Z - T_X f_\lambda) - (g - T f_\lambda) \right]. \end{aligned}$$

Therefore, Lemma A.5 and Lemma A.10 show that

$$\left\| T_\lambda^{-\frac{1}{2}} \left( \tilde{g}_Z - T_{X\lambda} f_\lambda \right) \right\|_{\mathcal{H}} = \left\| T_\lambda^{-\frac{1}{2}} \left[ (\tilde{g}_Z - T_X f_\lambda) - (g - T f_\lambda) \right] \right\|_{\mathcal{H}} = o_{\mathbb{P}}\left( n^{-\min(s,2)\theta/2} \right)$$

for both $s > \alpha_0$ and $s \leq \alpha_0$ cases. $\square$

## A.3 Approximation results

Let us further denote

$$\xi(x) = T_\lambda^{-\frac{1}{2}} (K_x f_\rho^*(x) - T_x f_\lambda). \tag{41}$$

Then, it is easy to check that

$$T_\lambda^{-\frac{1}{2}} \left[ (\tilde{g}_Z - T_X f_\lambda) - (g - T f_\lambda) \right] = \frac{1}{n} \sum_{i=1}^n \xi(x_i) - \mathbb{E}_{x \sim \mu} \xi(x).$$

The following lemma deals with the easy case when $s > \alpha_0$.

**Lemma A.5.** *Suppose Assumptions 1-3 hold and $s > \alpha_0$. Let $\lambda \asymp n^{-\theta}$ with $\theta \in (0, \beta)$ and $\delta \in (0, 1)$. Then, for $\alpha > \alpha_0 = \beta^{-1}$ being sufficiently close, it holds with probability at least $1 - \delta$ that*

$$\left\| T_\lambda^{-\frac{1}{2}} \left[ (\tilde{g}_Z - T_X f_\lambda) - (g - T f_\lambda) \right] \right\|_{\mathcal{H}} \leq C \ln \frac{2}{\delta} \cdot \left( M_\alpha^2 \frac{\lambda^{-\alpha}}{n} + M_\alpha \sqrt{\frac{\lambda^{-\alpha} \ln n}{n}} \right) \lambda^{\tilde{s}/2}, \quad (42)$$

*where $\tilde{s} = \min(s, 2)$. Consequently,*

$$\left\| T_\lambda^{-\frac{1}{2}} \left[ (\tilde{g}_Z - T_X f_\lambda) - (g - T f_\lambda) \right] \right\| = o_{\mathbb{P}}(\lambda^{\tilde{s}/2}) = o_{\mathbb{P}}(n^{-\tilde{s}\theta/2}). \quad (43)$$

Before proving Lemma A.5, we have to introduce the following proposition bounding the RKHS norm of the regularized basis function, which is a part of Li et al. [2023a, Corollary 5.6].

**Proposition A.6.** *Suppose $\mathcal{H}$ has embedding index $\alpha_0$. Then for any $\alpha > \alpha_0$,*

$$\left\| T_\lambda^{-1/2} k(x, \cdot) \right\|_{\mathcal{H}} \leq M_\alpha \lambda^{-\alpha/2}, \quad \mu\text{-a.e. } x \in \mathcal{X}. \quad (44)$$

*Proof of Lemma A.5.* To use Bernstein inequality in Lemma B.4, let us bound the $m$-th moment of $\xi(x)$:

$$\mathbb{E}\|\xi(x)\|_{\mathcal{H}}^m = \mathbb{E}\left\| T_\lambda^{-\frac{1}{2}} K_x (f_\rho^*(x) - f_\lambda(x)) \right\|_{\mathcal{H}}^m$$
$$\leq \mathbb{E}\left[ \left\| T_\lambda^{-\frac{1}{2}} k(x, \cdot) \right\|_{\mathcal{H}}^m \cdot \mathbb{E}\left( \left| f_\rho^*(x) - f_\lambda(x) \right|^m \mid x \right) \right]. \quad (45)$$

The first term in (45) is bounded through (44). For the second term, since $s > \alpha_0$, using the embedding condition and Lemma A.3, we have

$$\left\| f_\lambda - f_\rho^* \right\|_{L^\infty} \leq M_\alpha \left\| f_\lambda - f_\rho^* \right\|_{[\mathcal{H}]^\alpha} \leq C M_\alpha \lambda^{\min(s-\alpha, 2)/2} \leq C M_\alpha \lambda^{(\tilde{s}-\alpha)/2},$$

where we notice that $\min(s - \alpha, 2) = \min(s, 2 + \alpha) - \alpha \geq \tilde{s} - \alpha$ for the last inequality. Moreover, Lemma A.3 also implies

$$\mathbb{E}\left| f_\lambda(x) - f_\rho^*(x) \right|^2 = \left\| f_\lambda(x) - f_\rho^*(x) \right\|_{L^2}^2 \leq C \lambda^{\tilde{s}} \ln \frac{1}{\lambda} \leq C \lambda^{\tilde{s}} \ln n.$$

Plugging in these estimations in (45), we get

$$(45) \leq (M_\alpha \lambda^{-\alpha/2})^m \cdot \left\| f_\lambda - f_\rho^* \right\|_{L^\infty}^{m-2} \cdot \mathbb{E}\left| f_\lambda(x) - f_\rho^*(x) \right|^2$$
$$\leq (M_\alpha \lambda^{-\alpha/2})^m \cdot \left( C M_\alpha \lambda^{(\tilde{s}-\alpha)/2} \right)^{m-2} \cdot (C \lambda^{\tilde{s}} \ln n)$$
$$\leq \frac{1}{2} m! \left( C M_\alpha^2 \lambda^{\tilde{s}-\alpha} \ln n \right) \cdot \left( C M_\alpha^2 \lambda^{-\alpha+\tilde{s}/2} \right)^{m-2}. \quad (46)$$

The proof is then complete by Lemma B.4. $\qquad \square$

The case of $s \leq \alpha_0$ is more difficult. We will use the truncation technique introduced in Zhang et al. [2023a]. The following lemma can be proven similarly to Lemma A.3.

**Lemma A.7.** *Under Assumptions 1 and 3, for any $0 \leq \gamma < s + 2$, we have*

$$\|f_\lambda\|_{[\mathcal{H}]^\gamma}^2 \asymp \begin{cases} \lambda^{s-\gamma}, & s < \gamma; \\ \ln \frac{1}{\lambda}, & s = \gamma; \\ 1, & s > \gamma. \end{cases} \quad (47)$$

*Proof.* Simply notice that

$$\|f_\lambda\|_{[\mathcal{H}]^\gamma}^2 = \sum_{i=1}^\infty a_i^2 \frac{\lambda_i^2}{(\lambda_i + \lambda)^2} (\lambda_i^s i^{-1}) \lambda_i^{-\gamma} \asymp \sum_{i=1}^\infty \left( \frac{\lambda_i^p}{\lambda_i + \lambda} \right)^2 i^{-1},$$

where $p = (s + 2 - \gamma)/2$. Then we can apply Proposition B.2. $\qquad \square$

Then, we are able to show the following concentration result about the truncated $\xi_i$'s, whose proof resembles that of Lemma A.5.

**Lemma A.8.** *Suppose Assumptions 1-3 hold and $s \le \alpha_0$. Let $\lambda \asymp n^{-\theta}$ with $\theta \in (0, \beta)$ and $\delta \in (0, 1)$. For any $t > 0$, denote $\Omega_t = \{x \in \mathcal{X} : |f_\rho^*(x)| \le t\}$ and $\bar{\xi}(x) = \xi(x)\mathbf{1}_{\{x \in \Omega_t\}}$. Then, for $\alpha > \alpha_0 = \beta^{-1}$ being sufficiently close, it holds with probability at least $1 - \delta$ that*

$$\left\| \frac{1}{n} \sum_{i=1}^n \bar{\xi}(x_i) - \mathbb{E}\bar{\xi}(x) \right\| \le C \ln \frac{2}{\delta} \cdot \left[ \frac{M_\alpha}{n} \left( M_\alpha \lambda^{-\alpha} + t\lambda^{-\frac{\alpha+s}{2}} \right) + M_\alpha \sqrt{\frac{\lambda^{-\alpha} \ln n}{n}} \right] \lambda^{s/2}. \quad (48)$$

*Consequently, if $t \asymp n^l$ with $l < 1 - \frac{\alpha+s}{2}\theta$, we have*

$$\left\| \frac{1}{n} \sum_{i=1}^n \bar{\xi}(x_i) - \mathbb{E}\bar{\xi}(x) \right\| = o_{\mathbb{P}}(\lambda^{s/2}). \quad (49)$$

*Proof.* We follow the same routine of the proof of Lemma A.5 and obtain (45) with $\xi$ replaced with $\bar{\xi}$. The only difference is that we have to control

$$
\begin{aligned}
\left\| \mathbf{1}\{x \in \Omega_t\}(f_\lambda - f_\rho^*) \right\|_{L^\infty} &\le \|f_\lambda\|_{L^\infty} + \left\| \mathbf{1}\{x \in \Omega_t\}f_\rho^* \right\|_{L^\infty} \\
&\le M_\alpha\|f_\lambda\|_{[\mathcal{H}]^\alpha} + t \\
&\le CM_\alpha\lambda^{(s-\alpha)/2} + t,
\end{aligned}
$$

where we apply Lemma A.7 at the second inequality. Then, (46) changes to

$$\frac{1}{2}m! \left( CM_\alpha^2\lambda^{\tilde{s}-\alpha} \ln n \right) \cdot \left( CM_\alpha^2\lambda^{-\alpha+\tilde{s}/2} + M_\alpha\lambda^{-\alpha/2}t \right)^{m-2}$$

and the rest follows. $\square$

To bound the extra error terms caused by truncation, we have to use the following proposition about the $L^q$ embedding of the RKHS [Zhang et al., 2023a, Theorem 5].

**Proposition A.9.** Under Assumption 2, for any $0 < s \le \alpha_0$ and $\alpha > \alpha_0$, we have embedding

$$[\mathcal{H}]^s \hookrightarrow L^{q_s}(\mathcal{X}, \mathrm{d}\mu), \quad q_s = \frac{2\alpha}{\alpha - s}. \quad (50)$$

**Lemma A.10.** *Suppose Assumptions 1-3 hold and $s \le \alpha_0$. Let $\lambda \asymp n^{-\theta}$ with $\theta \in (0, \beta)$ and $\delta \in (0, 1)$. Then*

$$\left\| T_\lambda^{-\frac{1}{2}} \left[ (\tilde{g}_Z - T_X f_\lambda) - (g - Tf_\lambda) \right] \right\| = o_{\mathbb{P}}(\lambda^{s/2}) = o_{\mathbb{P}}(n^{-s\theta/2}). \quad (51)$$

*Proof.* We will choose $t = n^l$ for some $l$ that will be determined later and choose some $\alpha > \alpha_0$ being sufficiently close. Using the same notations as in (49), we decompose

$$
\begin{aligned}
\left\| \frac{1}{n} \sum_{i=1}^n \xi(x_i) - \mathbb{E}\xi(x) \right\|_{\mathcal{H}} &\le \left\| \frac{1}{n} \sum_{i=1}^n \bar{\xi}(x_i) - \mathbb{E}\bar{\xi}(x) \right\|_{\mathcal{H}} + \left\| \frac{1}{n} \sum_{i=1}^n \xi(x_i)\mathbf{1}_{\{x_i \notin \Omega_t\}} \right\|_{\mathcal{H}} \\
&\quad + \left\| \mathbb{E}\xi(x)\mathbf{1}_{\{x \notin \Omega_t\}} \right\|_{\mathcal{H}}.
\end{aligned} \quad (52)
$$

The first term in (49) is already bounded by (49) if $l < 1 - \frac{\alpha+s}{2}\theta$. To bound the second term in (52), we notice that

$$x_i \in \Omega_t, \ \forall i = 1, \ldots, n \quad \text{implies} \quad \frac{1}{n} \sum_{i=1}^n \xi(x_i)\mathbf{1}_{\{x_i \notin \Omega_t\}} = 0.$$

Since Markov's inequality yields

$$\mathbb{P}_{x\sim\mu}\{x \notin \Omega_t\} \le t^{-q}\|f_\rho^*\|_{L^q}^q, \quad (53)$$

where $q = \frac{2\alpha}{\alpha - s}$, we get

$$\mathbb{P}\{x_i \in \Omega_t, \ \forall i\} = (\mathbb{P}_{x \sim \mu}\{x \in \Omega_t\})^n = (1 - \mathbb{P}_{x \sim \mu}\{x \notin \Omega_t\})^n \geq (1 - t^{-q}\|f_\rho^*\|_{L^q}^q)^n.$$

So the second term vanishes with high probability as long as $l > 1/q$.

For the third term in (52), using (44), we get

$$
\begin{aligned}
\left\|\mathbb{E}\xi(x)\mathbf{1}_{\{x \notin \Omega_t\}}\right\|_{\mathcal{H}} &\leq \mathbb{E}\left\|\xi(x)\mathbf{1}_{\{x \notin \Omega_t\}}\right\|_{\mathcal{H}} \\
&= \mathbb{E}\left[\mathbf{1}_{\{x \notin \Omega_t\}}(f_\rho^*(x) - f_\lambda(x))\left\|(T_\lambda)^{-1/2}k(x, \cdot)\right\|_{\mathcal{H}}\right] \\
&\leq M_\alpha \lambda^{-\alpha/2}\mathbb{E}\left[\mathbf{1}_{\{x \notin \Omega_t\}}(f_\rho^*(x) - f_\lambda(x))\right] \\
&\leq M_\alpha \lambda^{-\alpha/2}\left[\mathbb{E}(f_\rho^*(x) - f_\lambda(x))^2\right]^{\frac{1}{2}}\left[\mathbb{P}\{x \notin \Omega_t\}\right]^{\frac{1}{2}} \\
&\leq M_\alpha \lambda^{-\alpha/2}\lambda^{s/2}t^{-q/2}\|f_\rho^*\|_{L^q}^{q/2}.
\end{aligned}
$$

Consequently, if $l > \frac{\alpha\theta}{q}$, then

$$\left\|\mathbb{E}\xi(x)\mathbf{1}_{\{x \notin \Omega_t\}}\right\|_{\mathcal{H}} = o(\lambda^{s/2}).$$

Finally, the three requirements of $l$ are

$$l < 1 - \frac{\alpha + s}{2}\theta, \quad l > \frac{1}{q}, \quad \text{and} \quad l > \frac{\theta\alpha}{q},$$

where $q = \frac{2\alpha}{\alpha - s}$. Since $\theta < \beta = \alpha_0^{-1}$, we can choose $\alpha$ sufficiently close to $\alpha_0$ such that $\theta\alpha < 1$. Then,

$$(1 - \frac{\alpha + s}{2}\theta) - \frac{1}{q} = (1 - \theta\alpha)\left(\frac{\alpha + s}{2\alpha}\right) > 0,$$

and thus

$$\frac{\theta\alpha}{q} < \frac{1}{q} < 1 - \frac{\alpha + s}{2}\theta,$$

showing that we can choose some $l$ satisfying all the requirements and the proof is finish. $\qquad \square$

### A.4 The noiseless case

The case when $\lambda = n^{-\theta}$ for $\theta < \beta$ is already covered in Theorem A.2. For the case $\theta \geq \beta$, the approximation Lemma B.5 no longer holds, and we must reduce it to the former case. However, there is no direct monotone property of $\mathbf{Bias}(\lambda)$. Nevertheless, we have the following monotone relation about the operator norms, whose proof utilizes the idea in Lin et al. [2021, Proposition 6.1] with modification.

**Proposition A.11.** Let $\psi_\lambda = \lambda(T_X + \lambda)^{-1} \in \mathscr{B}(\mathcal{H})$. Suppose $\lambda_1 \leq \lambda_2$, then for any $s, p \geq 0$,

$$\left\|T^s\psi_{\lambda_1}^p\right\|_{\mathscr{B}(\mathcal{H})} = \left\|\psi_{\lambda_1}^p T^s\right\|_{\mathscr{B}(\mathcal{H})} \leq \left\|T^s\psi_{\lambda_2}^p\right\|_{\mathscr{B}(\mathcal{H})} = \left\|\psi_{\lambda_2}^p T^s\right\|_{\mathscr{B}(\mathcal{H})}. \tag{54}$$

*Proof.* Let us denote by $\preceq$ the partial order induced by positive operators. Since the function $\lambda \mapsto \frac{\lambda}{z + \lambda}$ is monotone decreasing with $\lambda$, we obtain $\psi_{\lambda_1}^{2p} \preceq \psi_{\lambda_2}^{2p}$, which further implies

$$T^s\psi_{\lambda_1}^{2p}T^s \preceq T^s\psi_{\lambda_2}^{2p}T^s.$$

Then, since $\|A\|^2 = \|AA^*\|$, we have

$$\left\|T^s\psi_{\lambda_1}^p\right\|_{\mathscr{B}(\mathcal{H})}^2 = \left\|T^s\psi_{\lambda_1}^{2p}T^s\right\|_{\mathscr{B}(\mathcal{H})} \leq \left\|T^s\psi_{\lambda_2}^{2p}T^s\right\|_{\mathscr{B}(\mathcal{H})} = \left\|T^s\psi_{\lambda_2}^p\right\|_{\mathscr{B}(\mathcal{H})}^2,$$

and the equality in (54) is proven by $\|A\| = \|A^*\|$. $\qquad \square$

**Lemma A.12.** *Under Assumption 1,2,3, assume further $s > 1$. Suppose $\lambda \asymp n^{-\theta}$ for $\theta \geq \beta$. Then,*

$$\mathbf{Bias}^2(\lambda) = O_{\mathbb{P}}^{\mathrm{poly}}(n^{-\min(s,2)\beta}). \tag{55}$$

*Proof.* Since $f_\rho^*$ is given in (12) and $s > 1$, we have $f_\rho^* \in [\mathcal{H}]^t$ for $1 \le t < s$. In particular, $f_\rho^* \in \mathcal{H}$, so the bias term can also be written as

$$\mathbf{Bias}(\lambda) = \left\| \lambda(T_X + \lambda)^{-1} f_\rho^* \right\|_{L^2}. \tag{56}$$

Moreover, from the construction (8) of $[\mathcal{H}]^t$, we may assume $f_\rho^* = T^{t/2}g$ for some $g \in L^2$ with $\|g\|_{L^2} \le C$, and restrict further that $t \le 2$. Let $\tilde\lambda \asymp n^{-l}$ for $l \in (0, \beta)$. Then, using the same notation in Proposition A.11, we have

$$\begin{aligned}
\mathbf{Bias}(\lambda) = \left\| \psi_\lambda f_\rho^* \right\|_{L^2} &= \left\| T^{1/2} \psi_\lambda T^{\frac{t-1}{2}} \cdot T^{1/2} g \right\|_{\mathcal{H}} \\
&\le \left\| T^{1/2} \psi_\lambda T^{(t-1)/2} \right\| \cdot \left\| T^{1/2} g \right\|_{\mathcal{H}} \\
&\le C \left\| T^{1/2} \psi_\lambda^{1/2} \right\| \cdot \left\| \psi_\lambda^{1/2} T^{\frac{t-1}{2}} \right\| \\
&\le C \left\| T^{1/2} \psi_{\tilde\lambda}^{1/2} \right\| \cdot \left\| \psi_{\tilde\lambda}^{1/2} T^{\frac{t-1}{2}} \right\| \\
&\le C \left\| T^{1/2} \psi_{\tilde\lambda}^{1/2} \right\| \cdot \left\| \psi_{\tilde\lambda}^{(2-t)/2} \right\| \cdot \left\| \psi_{\tilde\lambda}^{\frac{t-1}{2}} T^{\frac{t-1}{2}} \right\| \\
&= C \tilde\lambda^{t/2} \left\| \psi_{\tilde\lambda}^{(2-t)/2} \right\| \cdot \left\| T^{1/2} T_{X\tilde\lambda}^{-1/2} \right\| \cdot \left\| T^{\frac{t-1}{2}} T_{X\tilde\lambda}^{-\frac{t-1}{2}} \right\| \\
&\le C \tilde\lambda^{t/2} \left\| T^{1/2} T_{X\tilde\lambda}^{-1/2} \right\|^t,
\end{aligned}$$

where we use Lemma B.6 for the last inequality. Finally, since $\tilde\lambda \asymp n^{-l}$ for $l < \beta$, Lemma B.5 implies that with high probability we have

$$\left\| T^{\frac{1}{2}} T_{X\tilde\lambda}^{-\frac{1}{2}} \right\| = \left\| T^{\frac{1}{2}} T_{\tilde\lambda}^{-\frac{1}{2}} T_{\tilde\lambda}^{\frac{1}{2}} T_{X\tilde\lambda}^{-\frac{1}{2}} \right\| \le \left\| T^{\frac{1}{2}} T_{\tilde\lambda}^{-\frac{1}{2}} \right\| \left\| T_{\tilde\lambda}^{\frac{1}{2}} T_{X\tilde\lambda}^{-\frac{1}{2}} \right\| \le 1 \cdot \sqrt{3} = \sqrt{3}.$$

Therefore, we obtain

$$\mathbf{Bias}(\lambda) = O_{\mathbb{P}}(\tilde\lambda^{t/2}) = O_{\mathbb{P}}(n^{-tl/2}).$$

Since $t < \min(s, 2)$ and $l < \beta$ can arbitrarily close, we conclude (55). $\square$

*Proof of Proposition* 4.4. Let us denote $\mathcal{F} = \left\{ f : \left\| f_\rho^* \right\|_{[\mathcal{H}]^s} \le R \right\}$ for convenience. Since $f_\rho^* \in \mathcal{H}$, the bias term can be given by

$$\mathbf{Bias}(\lambda) = \left\| f_\rho^* - T_X(T_X + \lambda)^{-1} f_\rho^* \right\|_{L^2} = \left\| (I - L_X) f_\rho^* \right\|_{L^2}$$

for a linear operator $L_X = T_X(T_X + \lambda)^{-1}$ on $\mathcal{H}$. Then,

$$\begin{aligned}
\sup_{f_\rho^* \in \mathcal{F}} \mathbf{Bias}(\lambda) = \sup_{f_\rho^* \in \mathcal{F}} \left\| (I - L_X) f_\rho^* \right\|_{L^2} &\overset{(a)}{=} \sup_{\|g\|_{\mathcal{H}} \le R} \left\| T^{\frac{1}{2}} (I - L_X) T^{\frac{s-1}{2}} g \right\|_{\mathcal{H}} \\
&= \sup_{\|g\|_{\mathcal{H}} \le R} \left\| (T^{\frac{s}{2}} - T^{\frac{s}{2}} L_X T^{\frac{s-1}{2}}) g \right\|_{\mathcal{H}} \\
&= \left\| T^{\frac{s}{2}} - T^{\frac{s}{2}} L_X T^{\frac{s-1}{2}} \right\|_{\mathscr{B}(\mathcal{H})} \\
&\overset{(b)}{\ge} \lambda_{n+1}^{s/2} = (n+1)^{-s\beta/2} = \Omega(n^{-s\beta/2}),
\end{aligned}$$

where in (a) we use the relation between the interpolation spaces and in (b) we use the fact that and $\|A - B\| \ge \lambda_{n+1}(A)$ for any operator $B$ with rank at most $n$ (see, for example, Simon [2015, Section 3.5]).

$\square$

# B  Auxiliary results

**Proposition B.1.** Let

$$f(z) = \frac{z^\alpha}{z + \lambda}.$$

Then,

(1) If $\alpha = 0$, then $f(z)$ is monotone decreasing.

(2) If $\alpha \in (0,1)$, then $f(z)$ is monotone increasing in $[0, \frac{\alpha\lambda}{1-\alpha}]$, and decreasing in $[\frac{\alpha\lambda}{1-\alpha}, +\infty)$. Consequently, $f(z) \leq \lambda^{\alpha-1}$.

(3) If $\alpha \geq 1$, then $f(z)$ monotone increasing on $[0, +\infty)$.

*Proof.* We simply notice that

$$f'(z) = \frac{z^{\alpha-1}}{(z+\lambda)^2}(\alpha\lambda - (1-\alpha)z).$$

$\square$

**Proposition B.2.** Suppose $c_\beta i^{-\beta} \leq \lambda_i \leq C_\beta i^{-\beta}$ and $p > 0$, then as $\lambda \to 0$, we have

$$\sum_{i=1}^{\infty}\left(\frac{\lambda_i^p}{\lambda_i+\lambda}\right)^2 i^{-1} \asymp \begin{cases} \lambda^{2(p-1)}, & p < 1; \\ \ln\frac{1}{\lambda}, & p = 1; \\ 1, & p > 1. \end{cases} \tag{57}$$

*Proof.* We first consider the case when $p < 1$. Since $c_\beta i^{-\beta} \leq \lambda_i \leq C_\beta i^{-\beta}$, from Proposition B.1, letting $q = \frac{p}{1-p}$, we have

$$\frac{\lambda_i^p}{\lambda_i+\lambda} \leq \begin{cases} \frac{C_\beta^p i^{-p\beta}}{C_\beta i^{-\beta}+\lambda}, & \text{if } C_\beta i^{-\beta} \leq q\lambda; \\ \lambda_i^p/\lambda_i \leq C_\beta^p i^{-(p-1)\beta}, & \text{if } C_\beta i^{-\beta} > q\lambda; \end{cases}$$

Therefore,

$$\sum_{i=1}^{\infty}\left(\frac{\lambda_i^p}{\lambda_i+\lambda}\right)^2 i^{-1} \leq C\sum_{i:C_\beta i^{-\beta}>q\lambda} i^{-2(p-1)\beta-1} + C\sum_{i:C_\beta i^{-\beta}\leq q\lambda}\frac{i^{-2p\beta}}{(C_\beta i^{-\beta}+\lambda)^2}i^{-1}$$

$$=: S_1 + S_2.$$

For $S_1$, noticing $C_\beta i^{-\beta} > q\lambda$ implies $i < (q\lambda/C_\beta)^{-1/\beta}$, we have

$$S_1 \leq C\sum_{i=1}^{\lfloor (q\lambda/C_\beta)^{-1/\beta}\rfloor} i^{-2(p-1)\beta-1} \leq C\lambda^{2(p-1)}.$$

For $S_2$, using Proposition B.1 again we have

$$S_2 \leq C\int_{(q\lambda/C_\beta)^{-1/\beta}-1}^{\infty}\frac{x^{-2p\beta}}{(C_\beta x^{-\beta}+\lambda)^2}x^{-1}\mathrm{d}x$$

$$= C\lambda^{2p-2}\int_{(C_\beta/q)^{1/\beta}-\lambda^{1/\beta}}^{\infty}\frac{y^{-2p\beta}}{(C_\beta y^{-\beta}+1)^2}y^{-1}\mathrm{d}y \quad (x = \lambda^{-1/\beta}y)$$

$$\leq C\lambda^{2p-2},$$

where we note that the last integral is bounded above by a constant. Therefore, we conclude that $\|f_\lambda - f_\rho^*\|_{[\mathcal{H}]^\gamma}^2 \leq C\lambda^{2p-2}$. For the lower bound, if $C_\beta i^{-\beta} \leq q\lambda$, we have

$$\frac{\lambda_i^p}{\lambda_i+\lambda} \geq \frac{c_\beta^p i^{-p\beta}}{c_\beta i^{-\beta}+\lambda},$$

and hence

$$\sum_{i=1}^{\infty}\left(\frac{\lambda_i^p}{\lambda_i+\lambda}\right)^2 i^{-1} \geq C\int_{(q\lambda/C_\beta)^{-1/\beta}}^{\infty}\frac{x^{-2p\beta}}{(C_\beta x^{-\beta}+\lambda)^2}x^{-1}\mathrm{d}x$$

$$= C\lambda^{2p-2}\int_{(C_\beta/p)^{1/\beta}}^{\infty}\frac{y^{-2p\beta}}{(C_\beta y^{-\beta}+1)^2}y^{-1}\mathrm{d}y$$

$$\geq C\lambda^{2p-2},$$

where we note that the last integral is independent of $\lambda$.

For the case $p = 1$, by Proposition B.1, we have

$$\sum_{i=1}^{\infty} \left( \frac{\lambda_i}{\lambda_i + \lambda} \right)^2 i^{-1} \leq C \sum_{i=1}^{\infty} \left( \frac{i^{-\beta}}{C_\beta i^{-\beta} + \lambda} \right)^2 i^{-1}$$

$$\leq C \sum_{i=1}^{\lfloor 2\lambda^{-1/\beta} \rfloor} \left( \frac{i^{-\beta}}{C_\beta i^{-\beta} + \lambda} \right)^2 i^{-1} + C \sum_{i=\lfloor 2\lambda^{-1/\beta} \rfloor+1}^{\infty} \left( \frac{i^{-\beta}}{C_\beta i^{-\beta} + \lambda} \right)^2 i^{-1}$$

$$\leq C \sum_{i=1}^{\lfloor 2\lambda^{-1/\beta} \rfloor} i^{-1} + C \int_{2\lambda^{-1/\beta}}^{\infty} \left( \frac{x^{-\beta}}{C_\beta x^{-\beta} + \lambda} \right)^2 x^{-1} \mathrm{d}x$$

$$\leq C \ln \frac{1}{\lambda} + C \int_{2}^{\infty} \left( \frac{y^{-\beta}}{C_\beta y^{-\beta} + 1} \right)^2 y^{-1} \mathrm{d}y$$

$$\leq C \ln \frac{1}{\lambda}.$$

For the lower bound, we have

$$\sum_{i=1}^{\infty} \left( \frac{\lambda_i}{\lambda_i + \lambda} \right)^2 i^{-1} \geq c \sum_{i=1}^{\lfloor \lambda^{-1/\beta} \rfloor} \left( \frac{i^{-\beta}}{c_\beta i^{-\beta} + \lambda} \right)^2 i^{-1}$$

$$\geq c \sum_{i=1}^{\lfloor \lambda^{-1/\beta} \rfloor} i^{-1} \geq c \ln \frac{1}{\lambda}.$$

For the case $p > 1$, by Proposition B.1, we have

$$\sum_{i=1}^{\infty} \left( \frac{\lambda_i^p}{\lambda_i + \lambda} \right)^2 i^{-1} \leq C \sum_{i=1}^{\infty} \frac{i^{-2p\beta}}{(C_\beta i^{-\beta} + \lambda)^2} i^{-1} \leq C \sum_{i=1}^{\infty} i^{-2(p-1)\beta-1} \leq C,$$

since the last series is summable. The lower bound is derived by

$$\sum_{i=1}^{\infty} \left( \frac{\lambda_i^p}{\lambda_i + \lambda} \right)^2 i^{-1} \geq \frac{\lambda_1^p}{\lambda_1 + \lambda} \geq c.$$

$\square$

**Proposition B.3.** Under Assumption 1, for any $p \geq 1$, we have

$$\mathcal{N}_p(\lambda) = \mathrm{tr} \left( T T_\lambda^{-1} \right)^p = \sum_{i=1}^{\infty} \left( \frac{\lambda_i}{\lambda + \lambda_i} \right)^p \asymp \lambda^{-1/\beta}. \tag{58}$$

*Proof.* Since $c\, i^{-\beta} \leq \lambda_i \leq C i^{-\beta}$, we have

$$\mathcal{N}_p(\lambda) = \sum_{i=1}^{\infty} \left( \frac{\lambda_i}{\lambda_i + \lambda} \right)^p \leq \sum_{i=1}^{\infty} \left( \frac{C i^{-\beta}}{C i^{-\beta} + \lambda} \right)^p = \sum_{i=1}^{\infty} \left( \frac{C}{C + \lambda i^\beta} \right)^p$$

$$\leq \int_{0}^{\infty} \left( \frac{C}{\lambda x^\beta + C} \right)^p \mathrm{d}x = \lambda^{-1/\beta} \int_{0}^{\infty} \left( \frac{C}{y^\beta + C} \right)^p \mathrm{d}y \leq \tilde{C} \lambda^{-1/\beta}.$$

for some constant $C$. The lower bound is similar. $\square$

The following inequality about vector-valued random variables is well-known in the literature [Caponnetto and De Vito, 2007].

**Lemma B.4.** *Let $H$ be a real separable Hilbert space. Let $\xi, \xi_1, \ldots, \xi_n$ be i.i.d. random variables taking values in $H$. Assume that*

$$\mathbb{E}\|\xi - \mathbb{E}\xi\|_H^m \leq \frac{1}{2} m! \sigma^2 L^{m-2}, \quad \forall m = 2, 3, \ldots. \tag{59}$$

*Then for fixed $\delta \in (0,1)$, one has*

$$\mathbb{P}\left\{ \left\| \frac{1}{n} \sum_{i=1}^n \xi_i - \mathbb{E}\xi \right\|_H \leq 2\left( \frac{L}{n} + \frac{\sigma}{\sqrt{n}} \right) \ln \frac{2}{\delta} \right\} \geq 1 - \delta. \tag{60}$$

*Particularly, a sufficient condition for (59) is*

$$\|\xi\|_H \leq \frac{L}{2} \ a.s., \ and \ \mathbb{E}\|\xi\|_H^2 \leq \sigma^2.$$

The following concentration result has been shown in Fischer and Steinwart [2020], Zhang et al. [2023a]. We use the form in Li et al. [2023a, Proposition 5.8] for convenience, see also Zhang et al. [2023a, Lemma 12].

**Lemma B.5.** *Suppose $\mathcal{H}$ has embedding index $\alpha_0$ and Assumption 1 holds. Let $\lambda = \lambda(n) \to 0$ satisfy $\lambda = \Omega\left(n^{-1/\alpha_0 + p}\right)$ for some $p > 0$ and fix arbitrary $\alpha \in (\alpha_0, \alpha_0 + p)$. Then, for all $\delta \in (0,1)$, when $n$ is sufficiently large, with probability at least $1 - \delta$,*

$$\left\| T_\lambda^{-\frac{1}{2}}(T - T_X)T_\lambda^{-\frac{1}{2}} \right\|_{\mathcal{H}} \leq C\left( \frac{\lambda^{-\alpha}}{n} \ln n \right)^{1/2}, \tag{61}$$

*where $C > 0$ is a constant no depending on $\delta, n, \alpha$, and we also have*

$$\left\| T_\lambda^{1/2} T_{X\lambda}^{-1/2} \right\|_{\mathscr{B}(\mathcal{H})}, \ \left\| T_\lambda^{-1/2} T_{X\lambda}^{1/2} \right\|_{\mathscr{B}(\mathcal{H})} \leq \sqrt{3}. \tag{62}$$

The following operator inequality[Fujii et al., 1993] will be used in our proofs.

**Lemma B.6** (Cordes' Inequality). *Let $A, B$ be two positive semi-definite bounded linear operators on separable Hilbert space $H$. Then*

$$\|A^s B^s\|_{\mathscr{B}(H)} \leq \|AB\|_{\mathscr{B}(H)}^s, \quad \forall s \in [0,1]. \tag{63}$$

The following lemma is a consequence of the fact that $x^r$ is operator monotone when $r \in (0,1]$ and is Lipschitz when $r > 1$, see Zhang et al. [2023a, Lemma 35] or Lin et al. [2018, Lemma 5.8].

**Lemma B.7.** *Suppose that $A$ and $B$ are two positive self-adjoint operators on some Hilbert space, then*

- *for $r \in (0,1]$, we have*

$$\|A^r - B^r\| \leq \|A - B\|^r.$$

- *for $r \geq 1$, denote $c = \max(\|A\|, \|B\|)$, we have*

$$\|A^r - B^r\| \leq r c^{r-1} \|A - B\|.$$

# C  Experiments

## C.1  Details of experiments in the main text

Recall that in the experiments section of the main text, we considered the kernel $k(x,y) = \min(x,y)$ and $x \sim \mathcal{U}[0,1]$. We know the eigensystem of $k$ that $\lambda_i = \left( \frac{2i-1}{2}\pi \right)^{-2}$ and $e_i(x) = \sqrt{2}\sin\left( \frac{2i-1}{2}\pi x \right)$. For the three target functions used in the experiments, simple calculation shows that the relative smoothness (source condition) of $\cos(2\pi x), \sin(2\pi x), \sin(\frac{3}{2}\pi x)$ are $0.5, 1.5, \infty$ respectively.

For some $f^*$, we generate data from the model $y = f^*(x) + \varepsilon$ where $\varepsilon \sim \mathcal{N}(0, 0.05)$ and perform KRR with $\lambda = cn^{-\theta}$ for different $\theta$'s with some fixed constant $c$. Then, we numerically compute the variance, bias and excess risk by Simpson's formula with $N \gg n$ nodes. Repeating the experiment for $n$ ranged in 1000 to 5000 with an increment of 100, we can estimate the convergence rate $r$ by a logarithmic least-squares $\log \text{err} = r \log n + b$ on the resulting values (variance, bias and excess risk). Figure 2 on page 22 shows the corresponding curves of the results in Table 1 in the main text. Note that for each setting, we tried different $c$'s in the regularization parameter $\lambda = cn^{-\theta}$ and show the curves under the best choice of $c$ ($c = 0.005$).

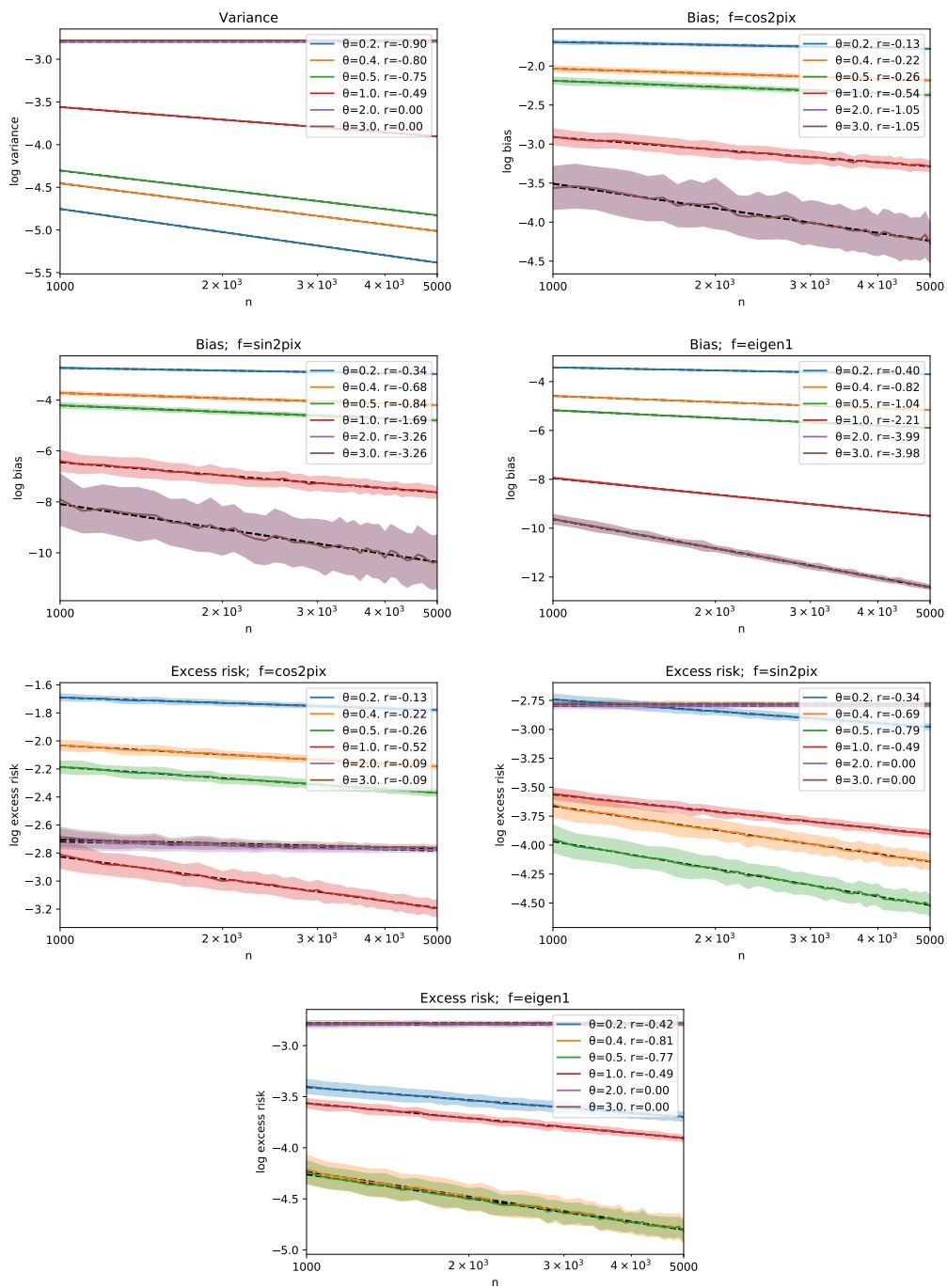

Figure 2: Decay curves of the variance; the bias and excess risk of three target functions. Both axes are logarithmic. The curves show the average bias over 100 trials; and the regions within one standard deviation are shown in the corresponding colors.

## C.2 Learning curves with different noises

Cui et al. [2021] discussed the 'crossover from the noiseless to noisy regime' and shown the interaction between the magnitude of noise and the sample size. As discussed in Remark 3.2 in the main text, our theory can also reflect this interaction. In Figure 3 on page 23, we exhibit the learning curves with different magnitudes of noises and visualize this interaction. Note that in the following the sample size is chosen as $10, 20, \cdots, 100, 120, \cdots, 1000, 1100, \cdots, 5000$, and we use the same kernel and data generation process as before. We repeat the experiments for 100 times for each sample size and present the average excess risk.

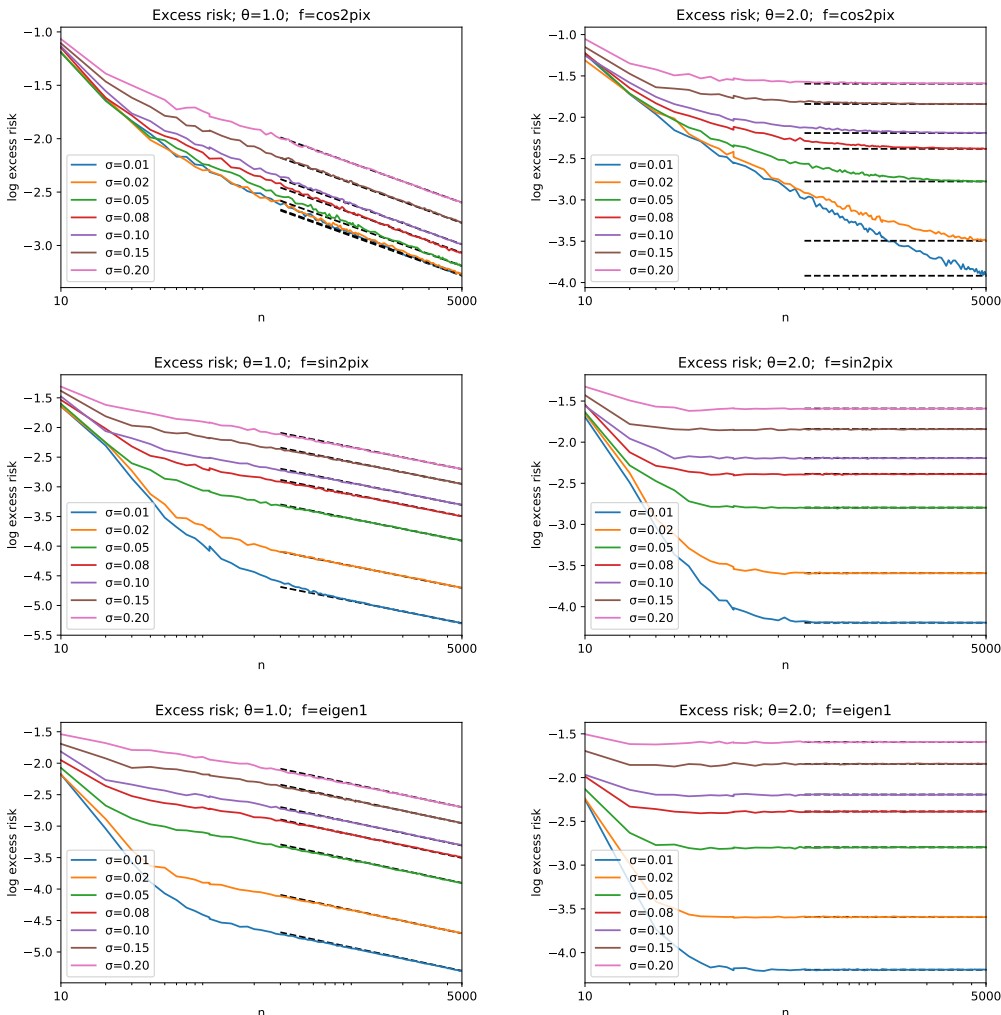

Figure 3: Learning curves of three target functions with different noises when choosing $\lambda = cn^{-\theta}$, $\theta = 1.0, 2.0$. Both axes are logarithmic. The black dashed lines represent the theoretical slopes under each choice of $\theta$.

In the above settings, the bias decays faster than variance. Figure 3 on page 23 shows that the excess risk decays fast when $n$ is relatively small and coincides the theoretical asymptotic rate in Theorem 3.2 when $n$ is large. The crossover happens for smaller $n$ when the magnitude of noise is larger. Similar phenomenon has also been reported by Cui et al. [2021, FIG.2, FIG.3]. In addition, comparing the sample size when crossover happens for three target functions, our results show that the crossover happens for smaller $n$ when the function is smoother, which is also consistent with Theorem 3.2.

Theorem 3.2 shows that when $\theta \geq \beta$, the excess risk is a constant asymptotically. Figure 4 on page 24 shows the curves of kernel interpolation ($\lambda = 0$). It can be seen that they are similar to the curves in the second column of Figure 3 on page 23, where we choose $\theta = \beta = 2$.

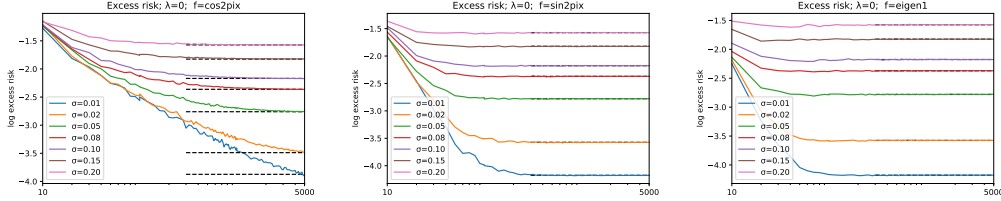

Figure 4: Learning curves of three target functions with different noises when choosing $\lambda = 0$. Both axes are logarithmic. The black dashed lines represent the theoretical slopes.

