# OpenReview forum: "On the Asymptotic Learning Curves of Kernel Ridge Regression under Power-law Decay"
_NeurIPS.cc/2023/Conference — NeurIPS 2023 poster_

### Official Review · Reviewer_3aFw · 2023-06-27

**Soundness:** 4 excellent
**Presentation:** 4 excellent
**Contribution:** 3 good
**Rating:** 7
**Confidence:** 3

**Summary:**

This paper provides a rigorous analysis of the asymptotic rates of kernel ridge regression. Building on works which analyzed the Gaussian design case, this work recovers the same rates using less severe restrictions on the distribution of features. The novel contribution here beyond similar works of Li et al 2023 (a,b) is to characterize the rate of the bias term. The authors derive a complete phase diagram of the possible behaviors and provide the optimal scaling of the ridge parameter $\lambda \sim n^{-\theta}$.  They provide experiments on simple sinusoidal regression problems where eigenvalues and eigenfunctions are exactly known and show that the theoretical rates match the experimental rates.

**Strengths:**

This paper gives a rigorous analysis of kernel regression rates when eigenvalues fall in a power law $\lambda_i \sim i^{-\beta}$. The specifically novel contribution in this work is obtaining the rate of decay of the bias term and giving an exact set of assumptions that give these rates (embedding index, source condition, Holder condition on kernel, etc).

The analysis recovers known (correct) results, which have been proven in the worst case, and calculated exactly under Gaussian design assumptions. The authors also provide empirical evidence to support their claimed rates on a toy data distribution, which is good for a theory paper.

**Weaknesses:**

**Obtained learning curves pre-exist in the literature**

The rates obtained match prior non-rigorous analyses and so do not exhibit *new* behaviors over what has been studied in the prior works. That said, the rigorous proof under relaxed set of conditions is still useful.

**Title seems too broad**

The title seems to cover too much territory. I would recommend a more descriptive title like "Rigorous analysis of bias and variance rates for KRR"

**Relevant works**

While the paper does mostly a good job of citing relevant prior works, it misses a few relevant works on this area

1. [Spigler et al 2019](https://arxiv.org/abs/1905.10843) which obtains rates in the noiseless regime using statistical physics techniques.
2. [Canatar et al 2021](https://www.nature.com/articles/s41467-021-23103-1), which also gives the bias and variance terms for noisy kernel ridge regression under the Gaussian design with the replica method.
3. [Simon et al 2023](https://openreview.net/forum?id=FDbQGCAViI), which uses a different heuristic to derive the average case KRR learning curve and introduces the idea of 'learnabilities' as a way of reasoning about the KRR predictor.

**Questions:**

1. Can similar analysis techniques be applied to other decay rates? Suppose the eigenvalues fell not as a power law but instead exponentially fast $\lambda_i = e^{- i \beta}$. Could the same techniques be used to prove rates in this case ? Another setting of interest are polynomial regression problems in high dimension such as those studied by [Canatar et al 2021](https://www.nature.com/articles/s41467-021-23103-1). Is a rigorous proof of these curves possible as well in the high dimensional limit?

2. The authors claim that "that the benign overfitting phenomenon may not exist if one trains a very wide neural network."  Which of the results in the paper do the authors thinks support this claim?

Typos
1. line 6 "attentions" should be "attention"
2. line 17 "arisen" -> "arising"
3. line 32  "attentions" -> "attention"
4. line 86 "orders" -> "order"
5. line 291 "mistery" -> "mystery"


**Limitations:**

The authors address the technical limitation of their current approach in the $\theta \geq \beta$ regime and leave the solution open as a future problem.

---

> ### Author Rebuttal · Authors · 2023-08-10
>
> Thanks for your positive feedback on our work.
>
> ### Weaknesses:
>
> > **Obtained learning curves pre-exist in the literature**
> >
> > The rates obtained match prior non-rigorous analyses and so do not exhibit *new* behaviors over what has been studied in the prior works. That said, the rigorous proof under relaxed set of conditions is still useful.
>
> **Our response:** Thanks for recognizing our contribution in providing a rigorous proof of prior works. To be honest, we owe much to the pioneer works by Bordelon et al. [2020] and Cui et al. [2021] who proposed the description of the exact order of the learning curve. It is our honor to have the chance to provide a rigorous proof of their inspirational statements.
>
> > **Title seems too broad**
> >
> > The title seems to cover too much territory. I would recommend a more descriptive title like "Rigorous analysis of bias and variance rates for KRR"
>
> **Our response:** The terminology "learning curve" was used in Bordelon et al. [2020] and Cui et al. [2021]. We think this title clearly indicates that our work is a follow-up work of their inspirational arguments/statements and we owe much credits to their works.  Thanks to your concern, we realized that we only solve the problem in the asymptotic sense under the power law decay, so how about we change the title to "On the asymptotic learning curve of KRR under power law decay"?
>
>
>
> > **Relevant works**
> >
> > While the paper does mostly a good job of citing relevant prior works, it misses a few relevant works on this area
>
> **Our response:** Thank you for referring these relevant works on this area. We will add them in the revision of the paper.
>
> ### Questions
>
> >  Can similar analysis techniques be applied to other decay rates? Suppose the eigenvalues fell not as a power law but instead exponentially fast-$\lambda_i=e^{-i\beta}$. Could the same techniques be used to prove rates in this case? Another setting of interest are polynomial regression problems in high dimension such as those studied by [Canatar et al 2021](https://www.nature.com/articles/s41467-021-23103-1). Is a rigorous proof of these curves possible as well in the high dimensional limit?
>
> **Our response:** We believe that our analysis techniques can be applied to other decay rates, including the exponential rate, but the analysis will be more sophisticated.
>
> Thanks for your nice reference. It greatly enlarges our understandings of kernel regression in high dimensions. We are also very interesting in determining the learning curve of KRR with high dimensional data. When the dimension $d$ is large, the eigenvalues may depend on $d$.  This makes the learning curve problem much more difficult.   Our current results can not be applied to the high dimensional data. We will try to extend our results to the high dimensional setting in future work.
>
>
>
> > The authors claim that "that the benign overfitting phenomenon may not exist if one trains a very wide neural network." Which of the results in the paper do the authors thinks support this claim?
>
> **Our response:** The theory of neural tangent kernel (NTK) shows that wide neural networks can be well approximated by certain kernel regression with the corresponding NTK (Jacot et al. 2018). On the other hand, our assumptions hold for NTK associated with ReLU neural networks as discussed in the paper, so our results imply that in the noisy case KRR with NTK does not perform well if $\lambda$ is relatively too small, which is the case when overfitting happens. Consequently, we can expect that overfitting in neural networks also impairs generalization.
>
> However, we note that there is a subtle difference between KRR and the kernel gradient descent regression which approximates wide neural networks, so our results could not directly apply to neural networks and we have to use the term "may" here. Nevertheless, we think that our theoretic results strongly suggest that the benign overfitting phenomenon for wide neural networks does not hold in our fix dimension setting.

---

> > ### Comment · Reviewer_3aFw · 2023-08-11
> > **Response to Authors**
> >
> > The authors gave very useful answers to my questions. I agree that the proposed title "On the asymptotic learning curve of KRR under power law decay" or something like it may be more descriptive of the contributions of the present work. The authors acknowledge that their results operate in fixed dimension but point out that the high dimensional limit could be an interesting future direction.
> >
> > I now understand the comment on benign overfitting in ReLU neural networks as a statement about the zero (or small) $\lambda$ limit and how it is not necessarily compatible with good generalization.  Based on these answers, I will increase my score.

---

### Official Review · Reviewer_xyck · 2023-07-03

**Soundness:** 3 good
**Presentation:** 2 fair
**Contribution:** 3 good
**Rating:** 6
**Confidence:** 3

**Summary:**

This paper analyzed generalization error rates for the Kernel Ridge Regression (KRR) problem. The authors make use of several generic assumptions, including (i) eigenvalue decay, (ii) Embedding index, (iii) source condition, and (iv) Holder continuity of the kernel. It is worth noting that assumptions (i), (ii), and (iv) are generally applicable to popular kernels. The findings of this study align with previous knowledge: in the absence of noise, interpolation achieves optimality, while in the presence of noise, a well-known Bias-Variance tradeoff emerges.

**Strengths:**

1. The paper is well-written, exhibiting a high level of clarity and coherence in its structure and language.
2. The main result presented in the paper is not only insightful but also highly intriguing, contributing valuable insights to the research area.
3. The authors have demonstrated diligent referencing and citation of relevant prior work, showcasing a thorough understanding of the existing literature in the field.

**Weaknesses:**

While I believe the paper is well-written, I feel that there are some areas where the authors assume background knowledge from the reader or make claims without providing adequate supporting discussions. I have identified a few specific points that I believe could benefit from clarification or additional explanation:

1. The contribution section appears to be highly technical. It would be helpful to provide a high-level, less technical overview of the contributions before diving into the specifics. For example, it would be beneficial to provide a brief explanation of the significance of variables such as β and s, as these variables are not defined prior to that section.

2. In line 47, it is mentioned that "our results...MAY suggest the benign...". The term "may" introduces uncertainty, but it is unclear how this conclusion was reached or what factors contribute to this interpretation/uncertainty. It seems that the authors may have intended to convey that since wide neural networks are essentially kernels, and the rates clearly demonstrate that noise impairs generalization, it is unlikely for the noise to have a benign effect. If this is indeed the case, it would be helpful to present this point in a clearer and less convoluted manner.

3. The second row of Figure 1 mentions "overfitting" and "underfitting," but there is no prior discussion or explanation of these terms in the paper. It would be beneficial to define these terms and provide some context.This would enhance the understanding of the results presented in the figure. Overall the second row figures were unclear to me.

4. In line 290, the statement mentions that "These results will help us better understand the generalization mistery of neural networks."
It is unclear how this conclusion was reached or what specific discussions led to it. It would be valuable to elaborate on the implications of the results and provide a more thorough explanation of how they contribute to a better understanding of the generalization of neural networks.(by the way "mistery" is a typo!)

Overall, by addressing these points, the paper can become more accessible to a broader range of readers and ensure a clearer understanding of the key concepts and conclusions.

**Questions:**

In addition to Weaknesses section I have Following questions,

I understand assumption 2 and 3 are common for these kind of analysis. But would you please elaborate,

1. What is the significance of the "embedding index"? What are the implications if α << 1/β or α >> 1/β?

2. What are the limitations of assumption 3? From my understanding, as "s" increases, the function f* becomes smoother since it belongs to H^s. Could you please clarify when this assumption holds or fails?


I am willing to increase my score if the authors address these questions in Weaknesses/Questions sections.

**Limitations:**

See weaknesses and questions.

---

> ### Author Rebuttal · Authors · 2023-08-10
>
> Thanks for your positive feedback on our work.
>
> ### Weaknesses:
>
> > *"1. The contribution section appears to be highly  ..."*
>
> **Our response:**
>
> We agree that our contribution is indeed relatively technical and thanks for your helpful suggestions.
>
> Following your advice, in the new revision we will add a high-level, less technical overview of our contributions and the explanation of the quantity $s$ and $\beta$ prior to that section as well. We plan to add several paragraphs discussing the significance of the parameters "$\beta$, $s$".  For example, we plan to include something like following into the revision:
>
> “The parameter $\beta$, describing the eigenvalue decay rate ($\lambda_i \asymp i^{-\beta}$, see Eq. (6)) of a certain operator, characterizes the span of the RKHS and the interplay between the RKHS and the marginal distribution of $\mu$, which is also referred to as the capacity condition or effective dimension condition (Caponnetto & De Vito, 2007; Steinwart et al., 2009). Moreover, larger $\beta$ implies better regularity of the functions in the RKHS.
>
> Another parameter $s$ (see Eq. (12)) further describes the relative smoothness of the regression function $f^*$ with respect to the RKHS. For example, $s=1$ implies that the regression function has no more smoothness rather than just belonging to the RKHS. Similarly, larger $s$ implies better relative smoothness of $f^*$. ”
>
> > *2. "In line 47, it is mentioned that "our results...MAY suggest the..". ..."*
> >
> > *4. "In line 290, the statement ..."*
>
> **Our response:** We are sorry that we did not make it clear enough and your explanation is indeed the case.
>
> The theory of neural tangent kernel (NTK) shows that wide neural networks trained by gradient descent can be well approximated by certain kernel regression called kernel gradient descent with the corresponding NTK (Jacot et al. 2018). In addition, kernel gradient descent with stopping time $t$ corresponds to KRR with regularization $\lambda = t^{-1}$ despite some subtle difference.
>
> On the other hand, our assumptions hold for NTK associated with ReLU neural networks as discussed in the paper, so our results imply that in the noisy case KRR with NTK does not perform well if $\lambda$ is not chosen properly.
>
> Consequently, we can expect that if the stopping time in training the neural network is not chosen properly, the neural network can not generalize well. In particular, too late stopping time, namely overfitting, will impair the generalization. This is the theoretical insight that we provide for understanding the generalization of neural networks.
>
> However, we note that there is a subtle difference between KRR and the kernel gradient descent regression which approximates wide neural networks, so our results could not directly apply to neural networks and we have to use the term "may" here. We are looking forward to dealing with this subtle difference rigorously in our future work.
>
> > *3. "The second row of Figure 1 mentions "overfitting" and "underfitting," ..."*
>
> **Our response:** We use the terms "overfitting" and "underfitting" in basically the traditional sense: if the variance dominates the bias, then overfitting happens, and if bias dominates the variance, underfitting happens. We will add more explanation on these terms in the new revision. The second row figures demonstrate convergence rates obtained by different magnitude of regularization ($\lambda = n^{-\theta}$). From the bias-variance tradeoff, we can expect that a relatively smaller $\lambda$ (i.e., larger $\theta$) results in overfitting that the variance dominates the bias, and a larger $\lambda$ leads to underfitting. Therefore, we can divide the domain of $\theta$  into regions of underfitting and overfitting (interpolating as an extreme case of overfitting). Moreover, since these regions are also related to the parameters $s$ and $\sigma^2$, we can make a phase diagram showing how these regions are affected by these parameters. The two plots show the impact of $s$ and $\sigma^2$ respectively. We hope this will clarify the plots.
>
> ## Questions:
>
> > *1. "What is the significance of the "embedding index"? ..."*
>
> **Our response:**
>
> If we did not mis-understand your question, the "$\alpha$" in your question should be $\alpha_{0}$, the embedding index.
>
> In our analysis, the embedding index serves as a critical tool to sharpen the concentration inequalities (e.g., Lemma 3.5 in the supplementary material).  Smaller $\alpha_{0}$ enables us to derive better (tighter) concentration inequalities.
>
> From the theory of RKHS, it can be shown that the embedding index $\alpha_0$ always satisfies $1/\beta \leq \alpha_0 \leq 1$. As we have listed in the paper, $\alpha_{0}=1/\beta$ holds for lots of RKHSs. Since we did not find any example with $\alpha_{0}>1/\beta$, we would like to further speculate that $\alpha_{0}=1/\beta$. To prove this conjecture would require more involved mathematical argument, so we  leave it to the future study.
>
> In case that $\alpha_{0}>1/\beta$ happens, we can only obtain the exact asymptotics for a restrictive range of $\lambda$ and the full learning curve can not be provided.
>
> > *2. "What are the limitations of assumption 3? ..."*
>
> **Our response:**
> You are right. The larger the $s$, the smoother the functions in $[\mathcal H]^{s}$.
>
> Assumption 3 assumes an expansion of the regression function under the eigen-basis such that the coefficients satisfy a polynomial decay. Since $\{e_i\}$ forms a orthonormal basis for $L^2$, any $L^2$ function admits an expansion over the basis with some square-summable coefficients. Then, the rate at which the coefficients tend to zero determines the smoothness index $s$, so this assumption is rather general. In fact, this assumption can also be weaken to other forms such as a polynomial decay of the tail sums of the coefficients, but we adopt this form of simplicity. Thus, we think this assumption is general enough to cover most the cases of interest.

---

> > ### Comment · Reviewer_xyck · 2023-08-19
> >
> > Thank you for your clarification; it has been received positively. I will adjust my score accordingly.

---

### Official Review · Reviewer_j73d · 2023-07-05

**Soundness:** 4 excellent
**Presentation:** 4 excellent
**Contribution:** 4 excellent
**Rating:** 8
**Confidence:** 3

**Summary:**

The authors focus on the learning curve of kernel ridge regression, an important topic in theoretical machine learning, which is also used to approximate the generalization ability of 'lazy trained/NKT regime' neural networks.

The authors provide a nearly full characterization of the learning curve in the setting of Source/Capacity conditions, considering the effects and interplay of the regularization parameter, the source condition, and the noise, rigorously closing many gaps in the literature, and in particular the picture arising from recent works in Neurips.

The paper's contributions include:
* Providing sharp estimates of the asymptotic orders of the bias term when the regularization parameter is not too small. This result holds for both well-specified and misspecified cases, improving the upper bounds given in previous research.
* Showing an upper bound of the bias term in the nearly interpolating case. This upper bound is tight and matches the information-theoretic lower bound.
* Providing learning curves of KRR for both noisy and noiseless cases, and the interplay between the two.



**Strengths:**

I believe this is a strong paper.

Most recent arguments on the learning curve of kernel methods were not entirely rigorous, and are based on the ‘Gaussian design’ assumption, which is certainly very restrictive. Even within the realm of Gaussians, Bordelon et al used a non-rigorous the replica method, while Cui et al used results from CGMT, but extrapolated their domain of validity. Random matrix arguments were also used by Jin et al, but some gap existed in the argument.

This paper bridges the mathematical gap and manages to prove rigorously a nearly complete picture, removing the Gaussian assumptions. It closes a clear gap in the mathematics of Kernel ridge regression.






**Weaknesses:**

I do not believe the paper has any particular weaknesses.


**Questions:**

* I would be interested to know more about the limitation of the predictions. Removing the Gaussian assumption is quite a feat, but are they any restrictions on the data structure except the power law distribution/embedding dimension? Indeed, it seems there are only Eigenvalue decay assumptions in the paper! Do this means that any distribution of data in the RKHS is OK and that only the Covariance matrix matters? It would help me immensely to be explicit and to give counter examples.

* With respect to Cui et al, it seems to me that the authors prove their entire phase diagram (Theorem 1 and 2), except for the left region of the blue region in Fig. 1 of Cui et al. I believe. The regime where θ ≥ β, is also very interesting, and I would be interested in understanding the reason why it is harder to control. Is it a possibility that in this region the data structure becomes relevant so that the Gaussian assumption is needed or do the authors believe it is only a technical problem? I remember a similar problem arises in Jacot et al.

**Limitations:**

From a theoretical point of view, the paper is certainly addressing its limitation and has no ethical issue that I can foresee. It would still be interesting to be given the code to reproduce the figure in the supplementary material.

---

> ### Author Rebuttal · Authors · 2023-08-10
>
> Thanks for your positive feedback on our work.
>
> Questions:
>
> > *1."I would be interested to know more about the limitation of the predictions. Removing the Gaussian assumption is quite a feat, but are they any restrictions on the data structure except the power law distribution/embedding dimension? Indeed, it seems there are only Eigenvalue decay assumptions in the paper! Do this means that any distribution of data in the RKHS is OK and that only the Covariance matrix matters? It would help me immensely to be explicit and to give counter examples."*
>
> **Our response:**
>
> The current work relies on two essential assumptions: 1.  the power law eigenvalue decay; and 2. the embedding index assumption.  These two assumptions implicitly depend on the data marginal distribution $\mu$ on $\mathcal{X}$, which serves as the underlying measure defining the eigenvalue (Eq. (4) and (5)).
>
> Since we have illustrted that the assumption 2 holds for many RKHSes, we would like to speculate that the embedding index assumption holds for all RKHS defined on $\mathcal X$ when $\dim \mathcal X$ is fixed. We are not sure if we can  construct a concrete example.
>
>
>
> > 2.*"With respect to Cui et al, it seems to me that the authors prove their entire phase diagram (Theorem 1 and 2), except for the left region of the blue region in Fig. 1 of Cui et al. I believe. The regime where θ ≥ β, is also very interesting, and I would be interested in understanding the reason why it is harder to control. Is it a possibility that in this region the data structure becomes relevant so that the Gaussian assumption is needed or do the authors believe it is only a technical problem? I remember a similar problem arises in Jacot et al."*
>
> **Our response:** We believe that the regime where $\theta \geq \beta$ is very tricky. In this regime, althought we can provide matching upper and lower bounds for the bias term, only the lower bound for the variance term is obtained so far.
>
> The main difficulty we are facing is that we can no longer use the concentration approach when $\theta \geq \beta$. The basic idea of concentration approach is that we want to show $(T_X+\lambda)^{-1}$ (for $T_X$, see Eq. 18) can approximate $(T+\lambda)^{-1}$ well. Since the eigenvalues of $T$ (and $T_X$ of course) tends to zero, $(T+\lambda)^{-1}$ diverges as $\lambda \to 0$ so we can not expect the concentration can still hold when $\lambda = n^{-\theta}$ is small, which turns out to fall into the regime $\theta \geq \beta$.
>
> Though our current technical tools fail in this regime, we expect that it is still be possible to rigorously prove the learning curve predicted by Bordelon et al. [2020], Cui et al. [2021] in this regime. We guess there might be some universality hidden in the learning curve problem of KRR. More precisely, if we take the learning curve as an output of the jont distribution of $\{\phi_{j}(x_{\mu})\}$, the results based on Gaussian design assumption (which assumes that the joint distribution is indpedeng Gaussian variables) and our results actually suggest that this output(the learning curve of KRR) does not depend on the distribution.

---

> > ### Comment · Reviewer_j73d · 2023-08-21
> >
> > Thank you for your detailed and instructive responses. I stand by my positive evaluation of the paper

---

### Official Review · Reviewer_oJ4i · 2023-07-06

**Soundness:** 3 good
**Presentation:** 3 good
**Contribution:** 1 poor
**Rating:** 6
**Confidence:** 3

**Summary:**

The paper studies the asymptotics of kernel ridge regression learning curves when the kernel eigenvalues and the ridge parameter both obey a power law.

**Strengths:**

1. The paper is readable and well written.

2. It derives asymptotic results for KRR for a specific choice of kernel eigenvalues and regularization.

3. The introduction of technical definitions is clear, and sufficient background for previous results is provided.

**Weaknesses:**

1. The claims of the paper are confusing to me. The double-descent phenomena (in this case sample-wise in the noisy case) implies that there is a second descent after the U-shaped curve, while the authors did not seem to reproduce that result.

2. The learning curves plotted in Figure 1. are with respect to the ridge parameter, while learning curves often refer to the generalization error as a function of training set size.

3. Experiments are not sufficient to demonstrate the tightness of the theoretical results.

4. The references (Bordelon et al. [2020], Cui et al. [2021]) cited in the paper demonstrate near perfect agreement with experimental learning curves despite the Gaussian assumption, but a discussion relating those results to the theory here is missing.

5. The claim regarding wide neural networks and benign overfitting stated in line 47 is neither explained nor demonstrated by experiments.

6. In general, a clear comparison to previous literature is missing, and it is hard to understand the novelty brought by this paper.

**Questions:**

1. In line 56, $s$ is not defined. Also, it does not match the main result (i.e. it should be $s \geq 2$  rather than $s \geq 1$).

2. In random matrix theory, precise asymptotics can be obtained by relying on Gaussian equivalence principles and provide almost perfect learning curves for kernel regression (see Cui et.al. 2021). What insights the theory developed here brings in addition to the previous results and why do you think Gaussian assumption works so well?

**Limitations:**

The limitations section is missing.

---

> ### Author Rebuttal · Authors · 2023-08-10
>
> Thanks for your carefully review.
>
> The KRR learning curve, recently proposed in Bordelon et al. [2020], Cui et al. [2021], aims to   depict the interaction between the source condition ($s$), capacity ($\beta$), the choice of the regularization parameter ($\lambda$), and the noise level ($\sigma^{2}$).The main novelty/contribution is that we provided the first mathematically solid solution under more realistic/general assumptions.
> ### Weakness:
> *"1. The claims of the paper are confusing..."*
>
>  There are several works used the terminology "double descent":
>
> 1.   [Nakkiran, 2019a](https://openreview.net/forum?id=B1g5sA4twr) used it to desribe the double descent behaviour as the number of parameters increases.
> 2.  ([Nakkiran, 2019b](http://arxiv.org/abs/1912.07242), [Hastie, 2020](http://arxiv.org/abs/1903.08560)) and related simple models used it to descirbe double descent behaviour as $\rho=\lim\frac{p}{n}$ increases.
> 3.  [Yilmaz, 2022](http://arxiv.org/abs/2206.01378) used it to describe that a second valley may occur after the U-shaped curve with respect to the regularization parameter in some certain settings.
>
> Since  the data of dimension is fixed in the current paper, the most relevant scenarios to us should be the 1st and the 3rd cases.
>
> If you are referring to the 1st scenario,  the double descent curve should be a curve describe the generalization performance of neural network along with the width $m$ goes to infinite. Our work only discuss the scenario $m=\infty$. We do not need to be consistent with the double descent phenomenon.
>
> If you are referring to the 3rd scenario, our results prove that the learning curve of KRR is exactly U-shaped in our setting, so the regularization-wise double-descent phenomenon does not exist in KRR.
>
> *"2.The learning curves plotted in Figure 1..."*
>
> In this paper, we adopt the terminology "learning curve" appeared in Bordelon et al. [2020], Cui et al. [2021] which refers to the generalization error curve with respect to the model complexity (corresponding to the ridge parameter $\lambda$ here). The learning curve is upper bounded traditionally by a U-shaped curve.
>
> *"3.Experiments are not sufficient to..."*
>
> The experiment is for illustrative purpose in a theoretical paper.  In Table 1, the experimental results of both variance and bias term match the theoretical values. Could you please provide more comments so that we could improve the experiments?
> *"5.The claim regarding wide neural networks and..."*
>
> Thanks for your suggestion. We are sorry that we did not make it clear enough in this version and will make more explanation in the new revision.
>
> 1. It is showed that  the generalization error of wide neural networks can be well approximated by that of kernel regression with NTK (see [Jacot et al. 2018](https://proceedings.neurips.cc/paper/2018/file/5a4be1fa34e62bb8a6ec6b91d2462f5a-Paper.pdf) and follow-up works).
> 2. Our results imply that in the noisy case KRR with NTK does not perform well if $\lambda$ is relatively too small, which is the case when overfitting happens.
> 3. Consequently, we can expect that overfitting in neural networks also impairs generalization.
>
> *"4.The references (Bordelon et al. [2020], Cui et al. [2021]) cited..."*
>
> *"6.In general, a clear comparison..."*
>
> We will make it is more clear in the revision. For example, we will add the following paragraph:
>
> "The learning curve of KRR was recently described by in  Bordelon et al. [2020], Cui et al. [2021]. However, these works are all based on the unrealistic Gaussian design assumption and some heuristic argument. Though the heuristic arguments are inspirational, a rigorous proof is indispensable if one plans to perform further investigations. In this work, we provide the first rigorous proof for most scenarios of the smoothness $s$, eigenvalue decary rate $\beta$, noise level $\sigma^{2}$ and the regularization parameter $\lambda$ based on the most common/realistic assumptions."
>
> We hope this will make the contribution and significance of our work more clear.
> ### Questions:
>
> *"1.In line 56, $s$ is not defined..."*
>
> We will add the explanation of the quantity $s$ before its appearance in the new revision: $s$ refers to the smoothness of the regression function (Eq. (12)). When $s \geq 1$, $f^* \in \mathcal{H}$ so this case is often called the well-specified case, and when $s \in (0,1)$, $f^* \notin \mathcal{H}$ so it is called the mis-specified case, so the statement in Line 56 is correct. Moreover, since our main results (Eq. (15) in Thm 3.1 and Eq. (16) in Thm 3.4) cover both the cases, we do not distinguish them in the statement.
>
> We guess the statement in Eq. (17) might be the reason for your confusion. In fact, Eq. (17) deals with the case $s > 1$ and the separation of $s \in (1,2]$ and $s > 2$ is due to technical difficulties. We hope these explanation will clarify the issue.
>
> *"2.In random matrix theory, precise asymptotics can be obtained..."*
>
> Thanks for your great question. It is actually a very delicate problem. Though solve it certainly beyond the scope of the current paper, we would like to discuss a little bit on it and will try to solve it in a future work.
>
> - (A) The Gaussian design assumption is an unrealistic assumption (i.e., one can not hope it hold for general kernel), however, with this assumption and some heuristic argument, Cui et al, 2021 can describe the learning curve in all scenarios.
>
> - (B) On the other hand, we provide a rigorous mathematical proof in almost all the scenarios via a completely different approach using the concentration inequality of integral operators.
>
> The interesting thing is that these two results actually suggests there might be a strong universality hidden here. This hypothetic universality might be the reason why Gaussian design assumption works so well. Furthermore, we think this universality (if we can make it rigorously) might help us extend our results from almost all the scenarios (A) to all the scenarios (B).

---

> > ### Comment · Reviewer_oJ4i · 2023-08-21
> >
> > I thank the authors for their detailed rebuttal and clarifications. With a more detailed survey of previous work and comparisons, I think it is a valuable work. I will adjust my score accordingly.

---

### Official Review · Reviewer_MHmF · 2023-07-12

**Soundness:** 2 fair
**Presentation:** 3 good
**Contribution:** 2 fair
**Rating:** 5
**Confidence:** 2

**Summary:**

This paper studies the generalization error/excess risk of the kernel ridge regression (KRR). Using the classical bias-variance decomposition, the authors gives bounds on the bias term, which is claimed to be tighter than previous literatures. Together with the variance term results from another paper  "Yicheng Li, Haobo Zhang, and Qian Lin. Kernel interpolation generalizes poorly, March 2023a.", the authors conclude the bounds on the excess risk asymptotic in the sample size, where the effect of the decay of the ridge is stated explicitly.

**Strengths:**

The paper gives a comprehensive reviews on the excess risk of KRR. There are experiments in both main text and appendix supporting the claimed result.

**Weaknesses:**

My biggest concern would be that this paper is merely an incremental paper from "Yicheng Li, Haobo Zhang, and Qian Lin. Kernel interpolation generalizes poorly, March 2023a."

According to the remarks 3.3 and 3.5, the main theorems 3.2 and 3.4 basically recover the results from previous works: "Andrea Caponnetto and Ernesto De Vito. Optimal rates for the regularized least-squares algorithm. Foundations of Computational Mathematics, 7(3):331–368, 2007. doi: 10.1007/ s10208-006-0196-8.", "Hugo Cui, Bruno Loureiro, Florent Krzakala, and Lenka Zdeborová. Generalization error rates in kernel regression: The crossover from the noiseless to noisy regime. Advances in Neural Information Processing Systems, 34:10131–10143, 2021." and "Yicheng Li, Haobo Zhang, and Qian Lin. Kernel interpolation generalizes poorly, March 2023a."  Other than tightening the bounds, I hardly see more novelty or improvement from the main result.

**Questions:**

Could you explain more on the novelty of the paper?

**Limitations:**

This is a theoretical paper, where the assumptions 1-5 are stated clearly. The authors has remarked that these assumptions hold in general settings.

---

> ### Author Rebuttal · Authors · 2023-08-09
>
> > My biggest concern would be that this paper is merely an incremental paper from "Yicheng Li, Haobo Zhang, and Qian Lin. Kernel interpolation generalizes poorly, March 2023a."
> >
> > According to the remarks 3.3 and 3.5, the main theorems 3.2 and 3.4 basically recover the results from previous works:
> > "Andrea Caponnetto and Ernesto De Vito. Optimal rates for the regularized least-squares algorithm. Foundations of Computational Mathematics, 7(3):331–368, 2007. doi: 10.1007/ s10208-006-0196-8.", "Hugo Cui, Bruno Loureiro, Florent Krzakala, and Lenka Zdeborová. Generalization error rates in kernel regression: The crossover from the noiseless to noisy regime. Advances in Neural Information Processing Systems, 34:10131–10143, 2021." and "Yicheng Li, Haobo Zhang, and Qian Lin. Kernel interpolation generalizes poorly, March 2023a." Other than tightening the bounds, I hardly see more novelty or improvement from the main result.
>
> ---
>
>
>
> The learning curves of KRR, depicting the interaction between the source condition ($s$) of the regression function, the capacity ($\beta$) of the underlying RKHS, the choice of the regularization parameter ($\lambda$), and the noise level ($\sigma^{2}$), might be one of the most important questions in the recent renaissance of the study of kernel methods.  It requires us to obtain the matching upper and lower bounds of the bias term, the variance term and thus the excess risk of KRR estimator for specific regression function $f^*$ with respect to the regularization parameter $\lambda$.
>
> We have to first emphasized that the learning curves is quite different than the traditional minimax theory:
>
> 1. the minimax theory only provide the worst-case lower bound for a class of regression functions, while the learning curves can show the exact rate for any specific regression function;
> 2. the minimax theory only considers the optimal choice of $\lambda$ and the optimal rate, while the learning curves determine the exact asymptotic rate ( both upper bound and lower bound) for any choice of $\lambda$ ( $\propto n^{?}$).
>
> We then list the differences in several related works (some of them already mentioned by you):
>
> 1. The work "Andrea Caponnetto and Ernesto De Vito. 2007" considered only the minimax framework. Since it explicitly assumed that $f\in [\mathcal H]^{s}$ for $1\leq s\leq 2$, it can not be applied to the more general situation where $f\in [\mathcal H]^{s}, s\in (0,2]$. Moreover, it did not provide the lower bounds for both the bias term and variance term, so it can not provide the learning curves.
> 2. The paper "Yicheng Li et al. 2023" focused on the variance term of the kernel interpolation, so it also can not answer the question of the learning curve.
> 3. "Hugo Cui et al. 2021"  firstly described the learning curve based on an heuristic argument and the Gaussian design assumption.  However, the Gaussian design assumption is unrealistic: it requires the eigenfunctions (see Eq. (5)) to be distributed independently, which can not be the case even for a simplest 1-d toy model (see the model in the Experiment section).
> 4. Random matrix arguments were used by [Jin et al](arxiv.org/abs/2110.12231), but some gaps existed in their argument (as discussed in [Zhang et al. 2023](https://openreview.net/forum?id=Kg2al3GXBR)).
>
> To sum up, several groups of researchers recently tried to solve the learning curve problem from various aspects. The novelty/contribution is that we provide the **first** mathematically solid solution of determining the exact asymptotic rates of learning curve in KRR under more realistic/general assumptions.
>
> We believe that our paper is more like a landmark paper in the theory of KRR rather than an incremental paper and hope that we can convince you of its importance to the community of kernel regression. Moreover, the learning curve of the KRR with data of fixed dimension would also be served as a prototype study of the investigation of the generalization ability of KRR with data of large dimension where $d$ may diverge.
>
> Lastly, to conclude this part, we want to talk a little bit more about the importance of determining the learning curve of KRR. Since Jacot et al. 2018 observed that the training process of the wide neural network trained by gradient descent can be well approximated by that of kernel regression with respect to NTK trained by gradient descent, studying the kernel regression with respect to the NTK becomes a common strategy to grasp the basic asymptotic properties of training a neural network.
>
> In addition, kernel gradient descent with stopping time $t$ roughly corresponds to KRR with regularization $\lambda = t^{-1}$ despite some subtle differences. Since it is of interest to understand how the condition of the regression function, the noise level and the choice of stopping time affect the generalization of neural networks, we want to investigate effect of these parameters on KRR, which is answered by the so-called learning curves.
>
> We hope that our explanation would address your concern and convince you that our contribution is enough.

---

> > ### Comment · Reviewer_MHmF · 2023-08-13
> >
> > Thank You for Your detailed comment. Now I have a better understanding on the novelty of the paper. I decide to raise my rating.

---

### Decision · Program_Chairs · 2023-09-21

**Decision:**

Accept (poster)

**Comment:**

The paper studies the generalization error/excess risk of the kernel ridge regression through a bias/variance decomposition. There is a vast literature on this topic in the past few years, and the paper does make a few novel contributions, mostly deriving a matching upper and lower bound (lower bounds are somewhat rare in the literature to the best of my knowledge) without the typical Gaussian assumption.

Overall, the reviewers are positive about the paper. Some concerns raised by the reviewers have been successfully answered by the authors during the discussion period. There are some concerns that the authors promised to address in the revised version. This includes better explanations regarding the novel aspects of the proof as well as changing the title (as raised by one reviewer, the current title is simple too general and does not clearly highlight the contribution of the paper). I think it would also be beneficial to make the abstract and introduction more precise regarding the setting of the paper (e.g. over-parametrized regime, polynomial decay for the eigenvalues of the kernel, etc). I trust the authors will address this point as promised in the rebuttal. I personally think this will significantly increase the impact of the paper as it otherwize might simply be overlooked due to the large literature in this field.

In conclusion, I recommend acceptance, and I expect the camera-ready version to address the remaining concerns.